# Comparative single-nucleus RNA-seq analysis revealed localized and cell type-specific pathways governing root-microbiome interactions

Qiuhua Yang[1,2], Zhuowen Li[1,2], Kaixiang Guan[1,2], Zhenghong Wang[1,2], Xianli Tang[1], Yechun Hong [1], Zhijian Liu [1], Jixian Zhai [1], Ancheng Huang [1]✉, Yanping Long [1]✉ & Yi Song [1]✉

Roots can recognize and differentially respond to beneficial and pathogenic microbes, which are fundamental for maintaining root microbiome homeostasis, plasticity, and plant fitness. Meanwhile, roots are highly heterogeneous tissues with complex cell-type compositions and spatially distinct developmental stages. We found that beneficial microbe specifically induces the expression of translation-related genes in the proximal meristem cells, and diverse ribosome proteins and translation regulators are necessary for beneficial microbe-mediated growth promotion. Notably, the root maturation zone can still mount localized immune responses to root pathogens, including genes related to camalexin and triterpene biosynthesis. A triterpene biosynthesis mutant blocked the microbiome reshaping process upon GMI1000 infection. Our results indicate roots may have specialized immune responses in the maturation zone, and provide important insights and vital resources for further elucidating regulators of root-microbe interactions and microbiome reshaping.

Plant roots are constitutively exposed to extremely complex soil microbial communities (microbiome), among which beneficial, pathogenic, and neutral microbes coexist[1]. Root-associated microbiomes significantly affect plant growth, immunity, and stress tolerance. Thus, plants have evolved sophisticated genetic mechanisms to shape a core microbiome[2], meanwhile, roots can also "cry for help" to selectively enrich beneficial microbes in response to certain stresses[3]. For instance, our previous study suggested that a receptor-like kinase gene, FERONIA, can selectively regulate beneficial *Pseudomonas* colonization in the natural microbiome[4]. To selectively regulate microbiome structure and maximize fitness, roots must precisely recognize and properly respond to microbes with different lifestyles.

Understanding how roots differentially respond to beneficial and pathogenic microbes would help reveal mechanisms underlying root-beneficial and root-pathogenic microbe interactions, thus pioneering strategies to maintain a healthy rhizosphere ecosystem in agriculture.

Plant immunity plays a pivotal role in maintaining microbiome homeostasis because loss of the immune system causes microbiome dysbiosis and cell damage[5]. Roots are surrounded by a much higher microbial load in the soil compared to leaves; thus, the regulation of the root immune system must be precisely controlled to avoid over-activation of immunity to soil microbiomes. To bypass root immunity, approximately 40% of microbiome members can suppress root immunity[6,7], and the interplay between root immunity and

[1]Shenzhen Key Laboratory of Plant Genetic Engineering and Molecular Design, Institute of Plant and Food Science, Department of Biology, School of Life Sciences, Southern University of Science and Technology, Shenzhen, China. [2]These authors contributed equally: Qiuhua Yang, Zhuowen Li, Kaixiang Guan, Zhenghong Wang. ✉e-mail: huangac@sustech.edu.cn; longyp@sustech.edu.cn; songy3@sustech.edu.cn

commensals affects both the composition of the root microbiome and the susceptibility of roots[6]. Roots can also positively downregulate immunity in response to beneficial *Pseudomonas*, largely dependent on the PSKR1 pathway[8]. Despite the importance of root immunity in root-microbe interactions, our understanding of root immune responses is much less than well-characterized leaf immune responses.

Plant roots are highly heterogeneous tissue containing different cell types and spatially distinct developmental stages. The most predominant feature of root immune responses is that roots show localized immune responsiveness that varies among different developmental zones and cell types[9,10]. Based on the bio-imaging analysis of specific immune marker genes using transgenic reporter lines, a previous study suggested mature (differentiated) outer cell layers lose the immune responsiveness to pure immune elicitors. Besides pure elicitors, roots also show mechanistically distinct immune responses in distinct root compartments to pathogens and commensals[11]. That suggests roots can recognize and respond to microbes with different lifestyles in a cell type-specific manner. However, bio-imaging and transgenic reporter-based systems have limited throughput and cannot be used to systematically dissect whole transcriptome responses in each cell type and developmental stage. To date, whether root maturation zones can effectively detect and show immune responses to different rhizosphere microbes remains unknown.

Recently developed single-cell RNA-seq technology enables us to profile transcriptional atlases at an unprecedented resolution. However, a big challenge for the application of scRNA-seq in plants is that it takes at least several hours to do protoplast isolation, and thus cannot profile real-time gene expression changes. However, plant immune responses to most elicitors can be detected within 30–90 min[12]. The aggressive mechanical shaking and enzymatic digestion during a long protoplasting period would inevitably cause unpredictable global transcriptional perturbation. To overcome this, we recently developed a protoplasting-free single-nucleus RNA-seq approach, which elegantly avoided perturbations during sample protoplasting[13]. Moreover, the fast development of single-cell (nucleus) RNA-seq studies thoroughly characterized cell-type-specific marker genes for *Arabidopsis* roots[14–17], which lays the foundations for comparative single-cell (nucleus) RNA-seq analysis during root-microbe interactions.

In this study, we utilized our recently developed protoplast-free single-nucleus RNA (snRNA-seq) sequencing approach to investigate the cell type and developmental stage-specific responses to beneficial and pathogenic rhizosphere microbes in roots[13]. We chose *Pseudomonas simiae* WCS417 and *Ralstonia solanacearum* GMI1000 as model beneficial and pathogenic microbes, respectively. WCS417 is a model beneficial microbe identified from a naturally occurring disease-suppressive soil that shows growth-promoting and disease-suppressive activities[18,19]. In contrast, *R. solanacearum* GMI1000 is one of the most devastating soil-borne bacterial pathogens infecting more than 250 plant species and causing serious agricultural losses worldwide[20]. We identified spatial and cell type-specific responses to beneficial and pathogenic microbes. Our work identified pathways related to root-beneficial and root-pathogenic microbe interactions and provided a wealth of data for further exploring root immune regulation and root-microbiome interactions.

## Results
### Establishing a single-cell atlas of root responses to rhizosphere microbes with different lifestyles
Here, we employed a 48-well plate-based hydroponic root-microbe interaction system[4,21] to segregate roots and leaves using a mesh and maintain roots in liquid media to facilitate interaction with rhizosphere bacteria (Fig. 1a). To investigate whether plant roots effectively recognize and differentially respond to GMI1000 and WCS417 within the initial few hours of interaction, we conducted qRT-PCR using whole excised roots and observed a modest upregulation of immune-responsive genes (*CYP71A12*, *MPK11*, *WRKY33*) at 6 h post-GMI1000 treatment (Supplementary Fig. 1). In contrast, WCS417 exhibited a weaker induction of these immune marker genes but displayed higher induction of *SRO4* (AT3G47720) (Supplementary Fig. 1), a gene responsive to WCS417 identified in a previous RNA-seq study[22]. We chose the 6-h time point for sampling due to the robust transcriptional response to both pathogens and beneficial microbes.

*Arabidopsis* roots were treated with either GMI1000, WCS417, or Mock (MgSO$_4$) for 6 h, and 12-day-old whole roots (5–7 cm) were harvested for snRNA-seq library preparation[13]. Two biological replicates were performed for each treatment. Following sequencing, data filtering, and quality control, we obtained 52,706 valid nuclei from the six libraries, encompassing 27,306 genes covering approximately 99.49% of the genome. The median gene count and UMI count for individual cells were 1001 and 1348, respectively (Supplementary Table 1). Integration of the six libraries was performed using Seurat CCA[23], and samples showed high consistency within each group (Supplementary Fig. 2a, b). For annotation purposes, CELLEX[24] was utilized to identify marker genes in each cluster (Supplementary Data 1), and their overlap with marker genes from a comprehensive *Arabidopsis* root snRNA-seq dataset was analyzed[25] (Supplementary Fig. 2c, d). Most clusters were assigned to corresponding cell types and developmental stages, with cluster 17 exhibiting mixed cell identities and further subdivided into 17_0 and 17_1 sub-clusters, resulting in a total of 27 clusters (Supplementary Fig. 2c, d). Leveraging cell type-specific markers identified in four published root snRNA-seq datasets[14–17], we annotated most clusters into 11 major root cell types, including proximal meristem (Cluster 2), root cap (Cluster 20), trichoblast (Cluster 21), atrichoblast (Cluster 11, 17_0, 0, 5), cortex (Cluster 19, 17_1, 1), endodermis (Cluster 14, 6, 15, 8, 24), pericycle (Cluster 3,16), procambium (Cluster 12, 4), phloem (Cluster 13), xylem (Cluster 25), and photosynthetic cells (Cluster 7, 9) (Fig. 1b, Supplementary Data 1). The expression patterns of well-characterized marker genes further validated the accuracy of the annotation[13–16,25,26] (Fig. 1c, Supplementary Table 2).

### SnRNA-seq profiling captures cell type-specific/enriched responses to WCS417 and GMI1000
To facilitate the comparison of gene expression patterns among major root cell types, we initially amalgamated all sub-cell types into 11 major root cell types (Supplementary Fig. 3a, b) and computed the number of differentially expressed genes (DEGs) in each major cell type. Both beneficial WCS417 and pathogenic GMI1000 triggered significant numbers of DEGs compared to the Mock group (Supplementary Fig. 3c, Supplementary Data 2). However, there were differing overlap ratios (ranging from 9.3% in the proximal meristem cells to 25.1% in xylem cells) between WCS417- and GMI1000-induced DEGs in different cell types (Supplementary Fig. 3d). The numbers of cells and the total detected genes are similar among different treatments (Supplementary Fig. 3e, f). Our data suggests that different root cell types can discern and differentially respond to beneficial and pathogenic microbes during the early stages (6 h) of interaction.

To validate our snRNA-seq dataset, we examined the expression patterns of several experimentally confirmed microbe-responsive marker genes. *Phytosulfokine receptor 1* (*PSKR1*) is a gene responsive to beneficial *Pseudomonas* in roots, acting as a negative regulator of root immunity and facilitating beneficial *Pseudomonas* colonization[8]. *Pseudomonas* colonization can induce the expression of a *pPSKR1: GUS* transgene reporter in roots[8]. Our data indeed revealed elevated expression of *PSKR1* in several cell types in response to WCS417, particularly in the cortex, endodermis, and procambium cells at this early stage (6 h) of WCS417 colonization (Supplementary Fig. 4a, Supplementary Data 3). We also checked the induction of two *PSKR1* ligand genes (*Phytosulfokine (PSK) 1* but not *PSK2*). We found that only *PSK1* was induced by beneficial WCS417 but not GMI1000 (Supplementary

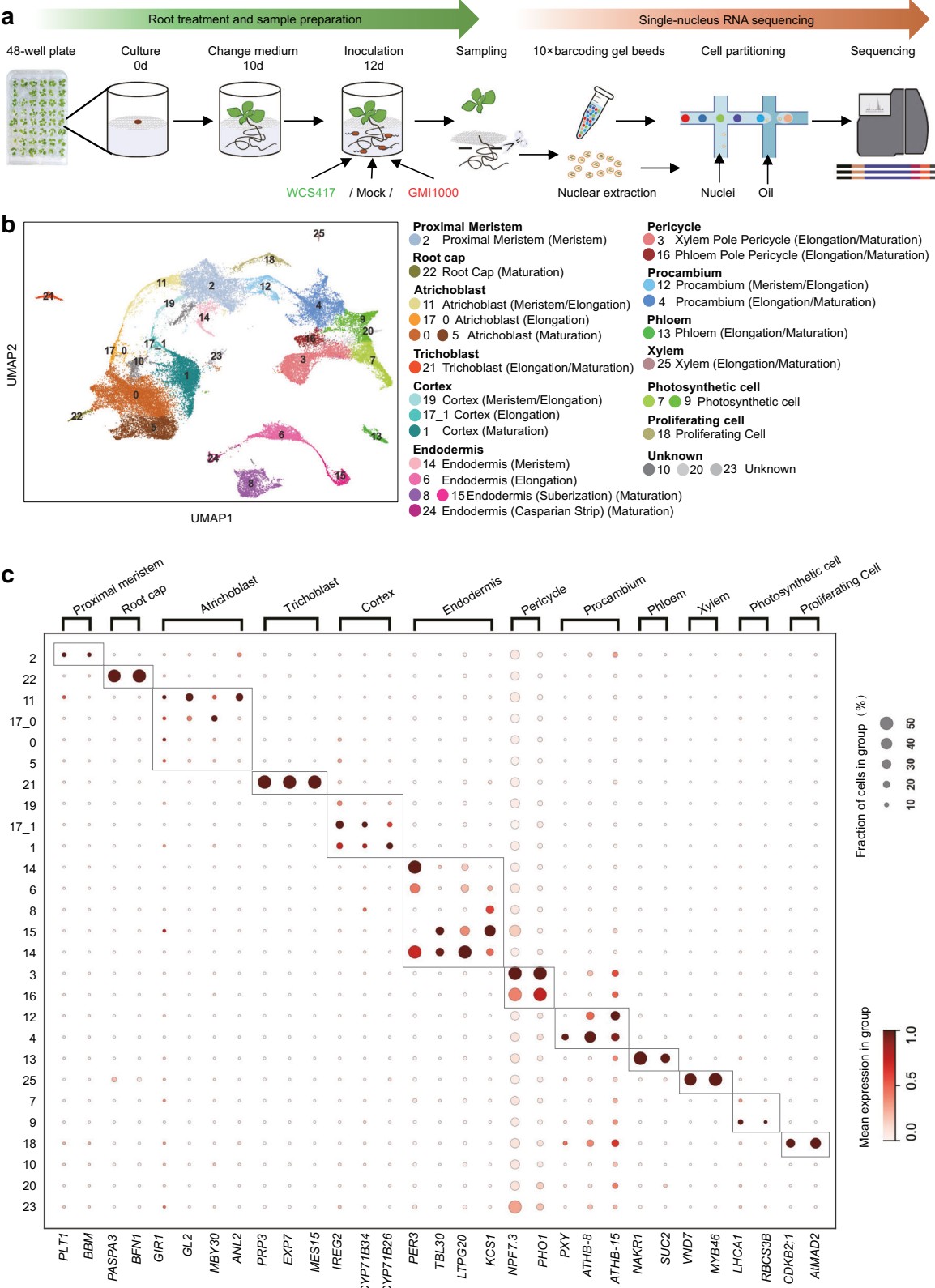

**Fig. 1 | Single-nucleus RNA-seq analysis of root responses to beneficial and pathogenic microbes. a** Schematic diagram showing the sample preparation and sequencing process for snRNA-seq. Plants were grown in 48-well plates with liquid MS media containing 2% sucrose for 10 days before changing to fresh liquid MS media without sucrose. Plant roots (12 days old) were then treated with 10 mM $MgSO_4$ (Mock), WCS417, or GMI1000 (final $OD_{600}$ = 0.05). Samples were harvested 6 h post-treatment. **b** UMAP visualization of 27 clusters in our snRNA-seq data of *Arabidopsis* whole roots. Combining with cell type-specific markers in recently published root scRNA-seq datasets, we annotated most clusters to corresponding cell type and developmental stages. **c** Dotplot showing the expression levels of previously reported cell type-specific marker genes in different cell clusters.

Fig. 4a). Furthermore, our data shows that *PSK1* exhibited enriched expression in the surface (atrichoblast and cortex) cell types, whereas *PSK2* was enriched in the inner (pericycle) cell types (Supplementary Fig. 4a), which is consistent with previous results based on promoter-based reporter lines[27]. Additionally, a recent study analyzed gene expression changes in different root cell types after 48 h of WCS417 inoculation using a fluorescence-activated cell sorting (FACS) approach[28]. They found that beneficial WCS417 could induce a subset of suberin biosynthesis (*GPAT7*) and transporter (*ABCG16*) genes in endodermis cells[28]. Our data confirmed that *GPAT7* and *ABCG16* are responsive to WCS417 and exhibited enriched expression in clusters of endodermis cells (Supplementary Fig. 4a). Notably, we not only observed consistent upregulation of suberin biosynthesis/transporter genes in response to beneficial WCS417 but also found that pathogenic GMI1000 induced even stronger upregulation of those genes (Supplementary Fig. 4a). This underscores the importance of conducting comparative analysis of beneficial and pathogenic microbe responses in roots to effectively distinguish a general non-self response[29] from a microbial lifestyle-specific response.

A previous cell-sorting-based RNAseq analysis found that trichoblasts exhibit higher basal and WCS417-induced expression of diverse defense genes relative to atrichoblasts[28]. However, most of their selected immune genes show higher basal-level expression in atrichoblast cells in our dataset, which contradicts their trend. This discrepancy might arise because our data capture early-stage (6 h) root-microbe responses (Supplementary Fig. 4b), which may differ from the 48-h responses in the previous study[28].

To investigate which biological processes are involved in the response to beneficial and/or pathogenic microbes, we conducted Gene Ontology (GO) enrichment analysis of WCS417- and GMI1000-induced DEGs (Fig. 2). We observed diverse shared GO terms enriched in both WCS417- and GMI1000-induced DEGs, indicating the presence of core microbial responses in roots[29]. For instance, salicylic acid (SA), phytoalexin, and toxin metabolism-related GO terms were highly enriched in most cell clusters following both GMI1000 and WCS417 treatments (Fig. 2a), consistent with the requirement of SA-related immune responses to prevent commensal overgrowth[8]. We also identified GO terms exclusively or mainly enriched only in WCS417- and GMI1000-induced genes (Fig. 2b, c), respectively. For WCS417-related GO responses, we observed diverse ribosome function, translation, and energy metabolism-related GO terms, particularly in proximal meristem cells (Fig. 2b). For GO terms mainly enriched in GMI1000-induced DEGs, "response to auxin" is enriched in the highest number of cell types (7 of 11 annotated cell types). This is consistent with the previous studies that GMI1000 may produce auxin-like molecules[30] and induce the expression of the *DR5* gene (a marker gene of auxin signaling)[31]. GMI1000 might induce the expression of auxin biosynthesis, signaling, and transport genes to manipulate root architecture upon infection[32]. Other GO terms like triterpenoid biosynthesis, immune responses, and phosphorelay signal transduction-related terms are also enriched in response to GMI1000 inoculation (Fig. 2c). Overall, our data indicate that roots exhibit both shared and distinct responses to beneficial and pathogenic microbes.

## Ribosome and translational regulators are necessary for WCS417-mediated root growth promotion

A myriad of plant-beneficial microbes, such as WCS417, have been observed to exert growth-promoting effects on their hosts, however, the molecular mechanisms underlying microbe-mediated growth promotion at the single-cellular level remain largely unexplored. To unravel potential pathways associated with WCS417-mediated growth promotion, we delved into our dataset. Initially, we observed that the root proximal meristem cells exhibited the lowest overlap ratio between WCS417 and GMI1000-induced DEGs (Supplementary Fig. 3d), indicating that root proximal meristem cells display the most

distinct responses to beneficial and pathogenic microbes. Furthermore, WCS417 triggered the second-highest number of DEGs in the proximal meristem cells (Supplementary Fig. 3c). Notably, GO enrichment analysis further suggested that various GO terms related to ribosome complex assembly, cellular respiration, and translational elongation were significantly enriched in DEGs from the proximal meristem cells only following WCS417 treatment, but not after GMI1000 treatment. Our findings suggest that these processes in proximal meristem cells may be linked to the growth-promoting effects mediated by beneficial WCS417.

Consequently, we analyzed the expression patterns of all WCS417-induced genes in proximal meristem cells (cluster 2) across all cell clusters (Fig. 3a). By employing k-means clustering (Fig. 3a), we identified 1055 genes in cluster-D that exhibited the highest induction in the proximal meristem cells only after WCS417 treatment, but not in response to GMI1000. These 1055 cluster D genes show significant enrichment of GO terms related to ribosome complex formation, translation elongation, oxidative phosphorylation, and ATP biosynthesis (Fig. 3b). This strongly indicates that those translation and energy metabolism processes might be related to beneficial microbe-induced growth-promoting effects. We identified a total of 249 ribosome function-related genes[33] in 1055 genes from cluster D. To validate their expression patterns in space, we used the root cell atlas website tool (https://rootcellatlas.org) to check gene expression patterns in a high-throughput way. This tool integrated diverse available public Arabidopsis root single-cell datasets[14–16,25,26,34,35] and provided high-confident gene expression patterns based on studies from different labs. We selected the top 53 highest expressed genes from those 249 ribosome-related genes (based on their basal expression levels in Col-0 root meristem cells), and found that they all show perfectly enriched expression patterns in meristem cells (Supplementary Fig. 5a). This further confirms the consistency and robustness of our data with previous single-cell studies in roots. Importantly, we also constructed a *pAT3G30600* (*RPS2*)::*YFP-NLS* reporter line, and experimentally validated the meristem cells enriched expression pattern of this ribosome function-related gene in space by confocal bio-imaging analysis (Supplementary Fig. 5b).

Ribosome function serves as a crucial bottleneck for protein translation, which is essential for development and stress responses[36]. Moreover, translational regulation broadly impacts both pattern-triggered and effector-triggered immunity in plants[37,38], among which *dst5* (HEM1, AT2G35110)[39] and *dst7* (CDC123, AT4G05440)[38] have been reported to be involved in translational regulation during effector-triggered immunity. HEM1 had been reported to be a master upstream hub regulator for organizing translation components, which interacts with 35 cytosolic ribosome translation initiation, elongation, or release factors[39]. Among those 35 HEM1-interacting ribosome proteins, 12 of them belong to the WCS417-induced cell type-specific DEGs in our cluster D (Supplementary Fig. 6). CDC123 regulates the assembly of the eukaryotic initiation factor 2 (eIF2) heterotrimer (α, β, and γ subunits) complex, which is central for translation initiation[38,40]. CDC123 protein can be immune-precipitated with 13 translation initiation factors[40], among them 6 genes belong to WCS417 specifically induced DEGs from cluster-D (Supplementary Fig. 6 and Fig. 3a, b). We found that both *cdc123* (*dst5*) and *hem1* (*dst7*) mutants could block the beneficial WCS417-mediated increase in lateral root numbers as well as shoot fresh weight (Fig. 3c–e). This confirms that immune-related translational regulators are also necessary for compatible root-beneficial WCS417 interactions and unveils a host pathway underlying beneficial WCS417-mediated growth promotion. We also measured the colonization levels of WCS417 and found that there was no significant difference between mutants and Col-0. This suggests that *cdc123*, *hem1* mutants triggered incompatible interaction with WCS417 is not due to impaired colonization, which could be due to downstream events related to translational regulation (Supplementary Fig. 7).

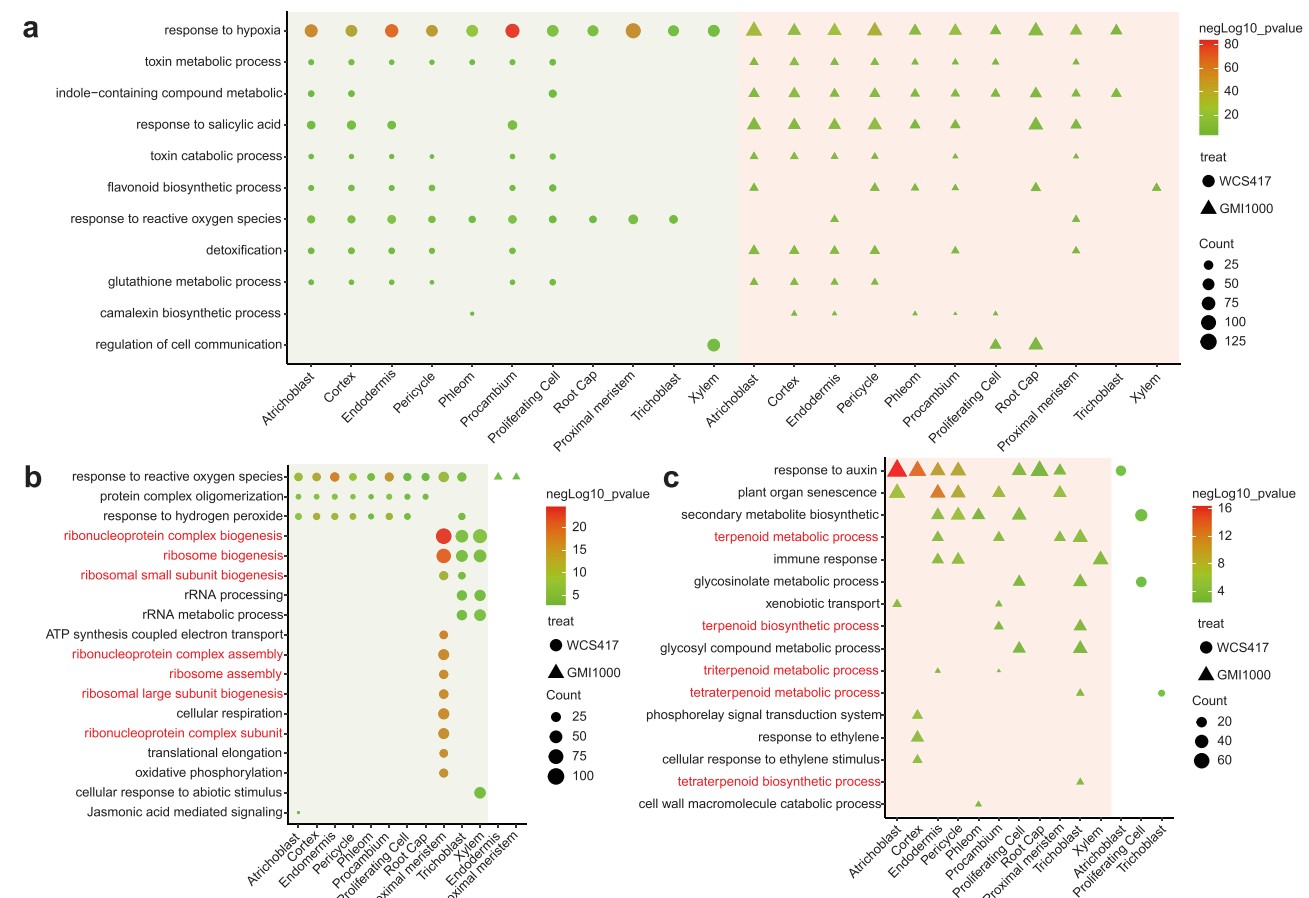

**Fig. 2 | GO enrichment analysis of WCS417 and GMI1000-induced DEGs from different cell types. a** Illustration of GO terms enriched both in WCS417 and GMI1000 triggered DEGs among different cell types. **b** GO terms mainly enriched in WCS417 upregulated genes in each cell type. **c** GO terms mainly enriched in GMI1000 upregulated genes in each cell type. The Y-axis represents the name of GO processes, while the X-axis shows different cell types. The color of the shape indicates the negative $\log_{10}$ (P-value) of GO terms. P-values were calculated using hypergeometric testing in the clusterProfiler package.

Although HEM1 and CDC123 are two master upstream translational regulators with diverse regulatory targets showing meristem cell enriched expression patterns (Supplementary Fig. 6), they do not show cell type-specific expression patterns or responsiveness to WCS417 (Supplementary Fig. 6b). It might be because those two master upstream translational regulators are mainly regulated at the post-transcriptional level. For example, CDC123 has an ATP grasping domain, and its translational regulating activity is gated by ATP concentration[40] (Supplementary Fig. 6a), and HEM1's activity is also regulated at the post-translational level by a phase separation mechanism to condensates with translation components[39] (Supplementary Fig. 6a). To further confirm whether WCS417 directly induced ribosome-related genes in the proximal meristem (cluster D genes) play an important role in WCS417-mediated plant growth promotion, we systematically tested 13 T-DNA insertion mutants corresponding to 13 genes belonging to ribosome assembly GO term in cluster D (Fig. 3f, g). Among 13 ribosome assembly-related mutants tested, six can also dampen or block WCS417-mediated growth promotion, including mutants of ribosome proteins RPL24 (AT3G53020), RPL4 (AT5G02870), RPS18A (AT1G22780), as well as translation elongation factor-like EF1B (AT5G19510), EF1β (AT1G30230) and ELF5A-2 (AT1G26630) (Fig. 3g). Collectively, our snRNA-seq data and genetic validations confirmed that WCS417 induces the expression of ribosome function-related genes, specifically in the proximal meristem, and that ribosome biogenesis and upstream translational regulators are necessary for WCS417-mediated plant growth promotion.

## GMI1000 triggers localized immune activation in the maturation cells

Plant roots exhibit cell-type and developmental stage-specific responses to various immune elicitors[9]. A significant finding is that mature atrichoblast cells largely lose their immune responsiveness to pure elicitors and can only mount immune responsiveness in the co-incidence of damage and elicitor[9]. However, whether the root maturation zone responds to live pathogens or beneficial microbe is unclear, especially from a cell type-specific transcriptome perspective. Given that we annotated diverse sub-cell clusters within several major root cell types (including atrichoblast, cortex, and endodermis, Fig. 1b), we aimed to investigate whether major root cell types exhibit sub-cluster (developmental stage)-specific expression of immune genes in response to rhizosphere microbes. A previous study compared the transcriptional landscape in response to diverse immune elicitors and characterized 970 DEGs generally induced by all tested immune elicitors[12]. Therefore, we employed these 970 genes[12] as core immune-responsive genes to examine whether different root cell types manifest distinct immune responses. Initially, we examined the number of core immune-responsive genes in DEGs from different sub-cell types (Fig. 4a–c). Intriguingly, we observed a gradual increase in GMI1000-responsive core immune genes and a gradual decrease in WCS417-responsive core immune genes in sub-clusters from the meristem to maturation zones in the cortex and atrichoblast cells (Fig. 4a–c), although this trend was less pronounced in the endodermis cells. This indicates that roots employ different immune-responsive

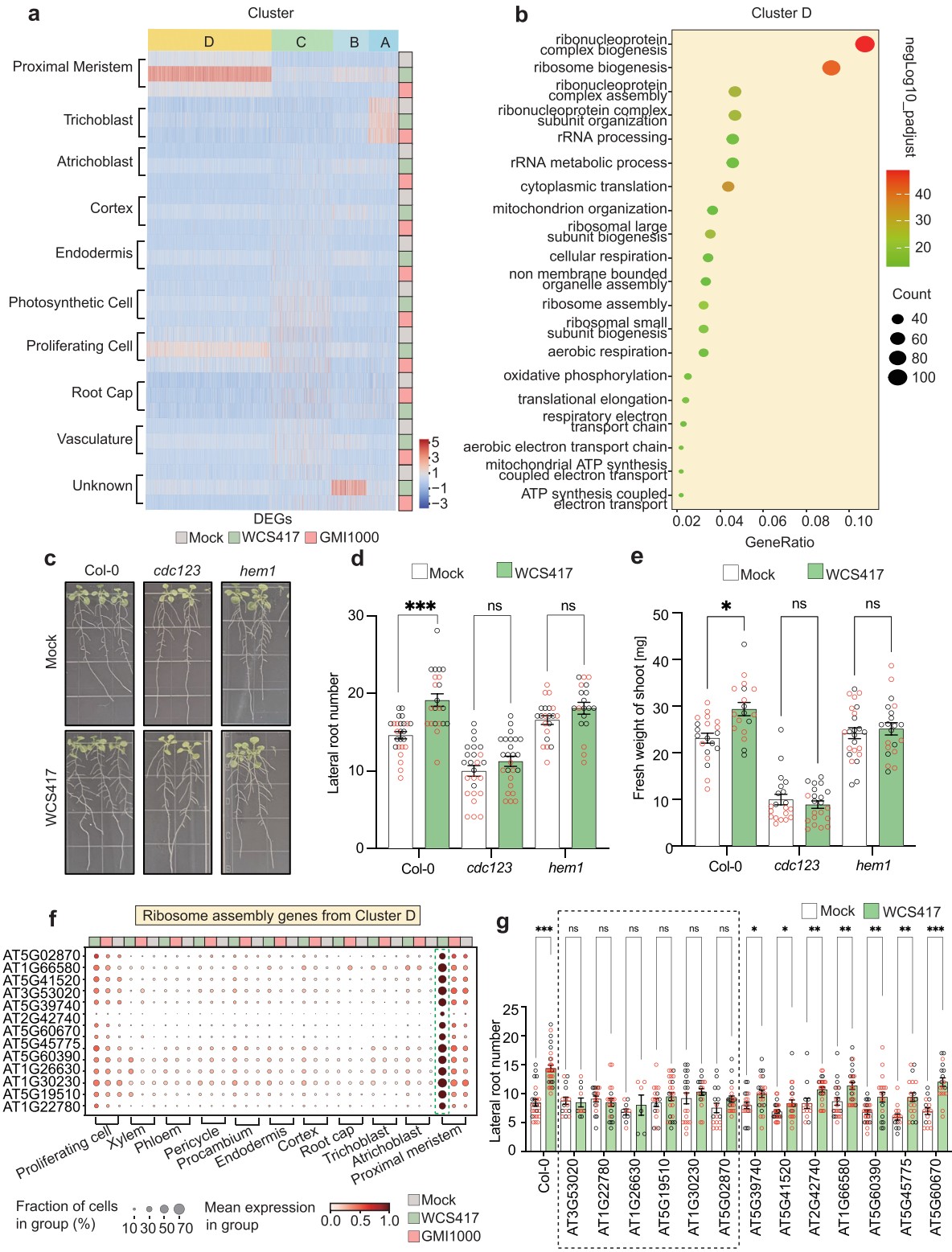

strategies for beneficial and pathogenic microbes. Surprisingly, we found that there are even more upregulated core immune genes after GMI1000 treatment in both mature atrichoblast (c0) and mature cortex (c1) cells than in younger cell types from the meristem (c11, c19) and elongation (c17_0, c17_1) zones. This suggests that the root maturation zone in atrichoblast and cortex cells still exhibits strong immune responses to the soil-borne pathogen GMI1000. Considering

that the root maturation zone only mounts immune in the co-incidence of damage and elicitor, we further tested whether GMI1000 induces cell damage or death in the maturation zone. We conducted propidium iodide (PI) based staining to detect the potential cell death (more permeable to PI) induced by GMI1000 (Supplementary Fig. 8)[32]. Consistent with the previous study[9], we did not find drastic cell death (broad PI signal inside the cell) in the maturation zone after 6 h of

**Fig. 3 | Ribosome and translation-related genes are necessary for beneficial WCS417-mediated growth promotion. a** K-means clustering analysis of the expression patterns of all WCS417-induced genes in the meristem cells (cluster 2). The heatmap is colored by the Z-score normalized CPM value. The annotation color bars on the right represent different treatments. **b** GO enrichment analysis of DEGs in cluster D, which shows strong upregulation induced by WCS417 treatment in cluster 2 cells. The size and color of the dot indicate enriched gene counts in the corresponding GO term and negative $\log_{10}$ (*P* adj), respectively. Adjusted *P*-values were calculated using the clusterProfiler package with Benjamini-Hochberg (BH) correction. **c** Mutants of known translational regulators block WCS417-mediated growth-promoting effect. **d, e** Quantification of lateral root numbers (**d**) in each

root and fresh weight of shoot (**e**) with and without WCS417 inoculation. **f** The expression levels of the 13 ribosome assembly-related genes from ribosome assembly GO term (**b**) in cluster D; **g** Quantification of lateral root numbers in 13 ribosome assembly-related mutants with and without WCS417 inoculation. The mutant names are listed above the corresponding genes. Results are presented as mean ± standard error of the mean (SEM). Asterisks represent the significant (*P < 0.05; **P < 0.01; ***P < 0.001, two-sided Student's *T*-test) differences between the WCS417-inoculated group and the MgSO4 group (Mock). The experiment was repeated twice, and different colors indicate data points from different experiments. For (**d**, **e**, and **g**), specific sample numbers are provided in the Source Data.

GMI1000 inoculation (Supplementary Fig. 8). How GMI1000 induces immune activation in the maturation zone during the first 6 h still requires more exploration.

A k-means clustering and GO enrichment analysis further unveiled complex expression patterns of those core immune-responsive genes in different cell types. Among these, some antimicrobial compound biosynthesis-related GO terms were routinely enriched in genes showing high expression in root maturation cell types (Supplementary Fig. 9). This indicates that antimicrobial metabolite-related immune genes might be specifically induced in root maturation cells. Plant secondary metabolites play pivotal roles in immune responses, among which tryptophan-derived indole-glucosinolates (IGSs) and camalexin are extensively studied antimicrobial metabolites[41]. Hence, we systematically examined whether IGS and camalexin biosynthesis pathways exhibit spatial (developmental stage-specific) expression patterns in atrichoblast, cortex, and endodermis cells. Most of these genes showed robust upregulation in response to GMI1000 in the maturation cells, particularly in atrichoblast and cortex cells (Fig. 4d–g). We observed that most upstream tryptophan metabolic genes, including *TSA1/TSB1*, *ASA1/ASB1*, *IGPS*, *PAT*, and *CYP79B2*, exhibited enriched induction in maturation cells. Moreover, the critical camalexin biosynthesis gene *CYP71A12* demonstrated very specific induction in the maturation zone of all three cell types (atrichoblast, cortex, and endodermis) upon GMI1000 treatment. We further conducted a comprehensive analysis of the IGS pathway genes in our atlas and confirmed IGS pathway shows enriched expression in cortex cells and an unknown cell type (Supplementary Fig. 10). We selected *CYP71A12* to experimentally validate its developmental stage-specific responsive patterns by constructing a *pCYP71A12::YFP-NLS* reporter line. Consistent with our snRNA-seq profiling, we observed a much stronger expression of the *pCYP71A12::YFP-NLS* signal in the maturation zone compared to the very weak expression in the meristem zone (Fig. 4h, i). The expression pattern of *CYP71A12* suggested that the unknown cell type 10 contains mixed cells from cortex and atrichoblast cells, which are hard to be clearly annotated. To further explore whether our observed expression pattern of the IGS pathway is correlated with canonic immune responses, we checked the expression of general immune or salicylic acid (SA) related immune marker genes. We found that some immune marker genes show similar expression patterns to *CYP71A12* (Fig. 4e–g), including *PBS3*, *MPK11*, *CBP60g*, and *SID2* show similar expression patterns (strong induction in response to GMI1000 in the maturation zone of atrichoblast and endodermis cells, Fig. 4e–g). Collectively, our data suggest that plant roots can exhibit sub-cell type (developmental stage)-specific responses to root pathogens and demonstrate that root maturation cells can mount strong immune responsiveness to GMI1000 during several hours of early-stage interactions.

## Localized expression of triterpene biosynthetic genes in response to GMI1000

Since our investigation revealed that root maturation cells exhibit immune responses to pathogens, particularly for genes related to

secondary metabolites, we sought to explore whether other secondary metabolites-related genes are also enriched in these maturation cells. We noted that terpenoid biosynthesis-related GO terms were exclusively enriched in GMI1000-treated roots (Fig. 2c). Interestingly, four thalianin biosynthesis genes exhibited very specific and robust induction in the maturation cell types across all three cell types (Fig. 5a–f). Furthermore, the expression levels of these genes were higher in the cortex and atrichoblast cells compared to endodermis cells (Fig. 5b). By generating a *pTHAS1*-driven YFP reporter line, we confirmed the enriched expression of *THAS1* in the maturation cortex and atrichoblast cells (Fig. 5c). Moreover, we also found that *pTHAS1::YFP-NLS* show much stronger induction in response to GMI1000 in the maturation zone compared to elongation and meristem zone (Fig. 5g, h). This perfectly validated the maturation cell-specific responsiveness of *THAS1 to* GMI1000 infection from our dataset (Fig. 5d–f). It is noteworthy that the enrichment of terpenoid biosynthesis-related GO terms was not identified in a previous time-series (0–96 h) bulk root RNA-seq dataset[32], possibly because gene expression changes of triterpenoid biosynthetic genes in a few cell types could not be effectively detected in bulk root RNA-seq studies. This further underscores the capability of the snRNA-seq approach in enhancing the resolution of discovering cell type-specific pathways.

## The triterpene biosynthetic pathway is required for reshaping the root microbiome upon *Ralstonia* infection

Triterpenes represent plant-specialized metabolites with selective antimicrobial activities and play crucial roles in shaping the *Arabidopsis* core root microbiome[42]. However, the biological context in which plants produce triterpenes to sculpt the microbiome remains elusive. The previous study identified 494 core OTUs (Operational Taxonomic Units) specifically enriched in the root microbiome of *Arabidopsis* compared with rice and wheat (neither of which produces triterpene products such as thalianin and arabidin). Remarkably, approximately one-third of these *Arabidopsis* specifically enriched OTUs were depleted in triterpene mutant lines, particularly in the *thas1* mutant defective in thalianin biosynthesis[42]. This suggests a crucial role of THAS1 in enriching *Arabidopsis*-specific root bacteria rather than repelling them[42]. Given our finding that triterpene biosynthesis-related genes were significantly induced by GMI1000, we hypothesized that roots might employ the THAS1 metabolic pathway to sculpt the microbiome in response to pathogens. To genetically confirm this, we conducted microbiome sequencing in Col-0 (wild type) and *thas1* mutant plants with/without GMI1000 inoculation in natural soil (Fig. 6a).

A total of 4,318,924 raw sequencing reads were obtained from all samples. Principal coordinated analysis (PCoA) based on Bray–Curtis distance revealed significant separation of samples by genotypes, thus validating the role of THAS1 in shaping root microbiome structure (Fig. 6b). Furthermore, we observed that *Ralstonia* GMI1000 infection significantly altered the microbiome structure in wild-type plants (Fig. 6c). Notably, roots failed to significantly alter the microbiome structure in response to GMI1000 infection in the *thas1* mutant

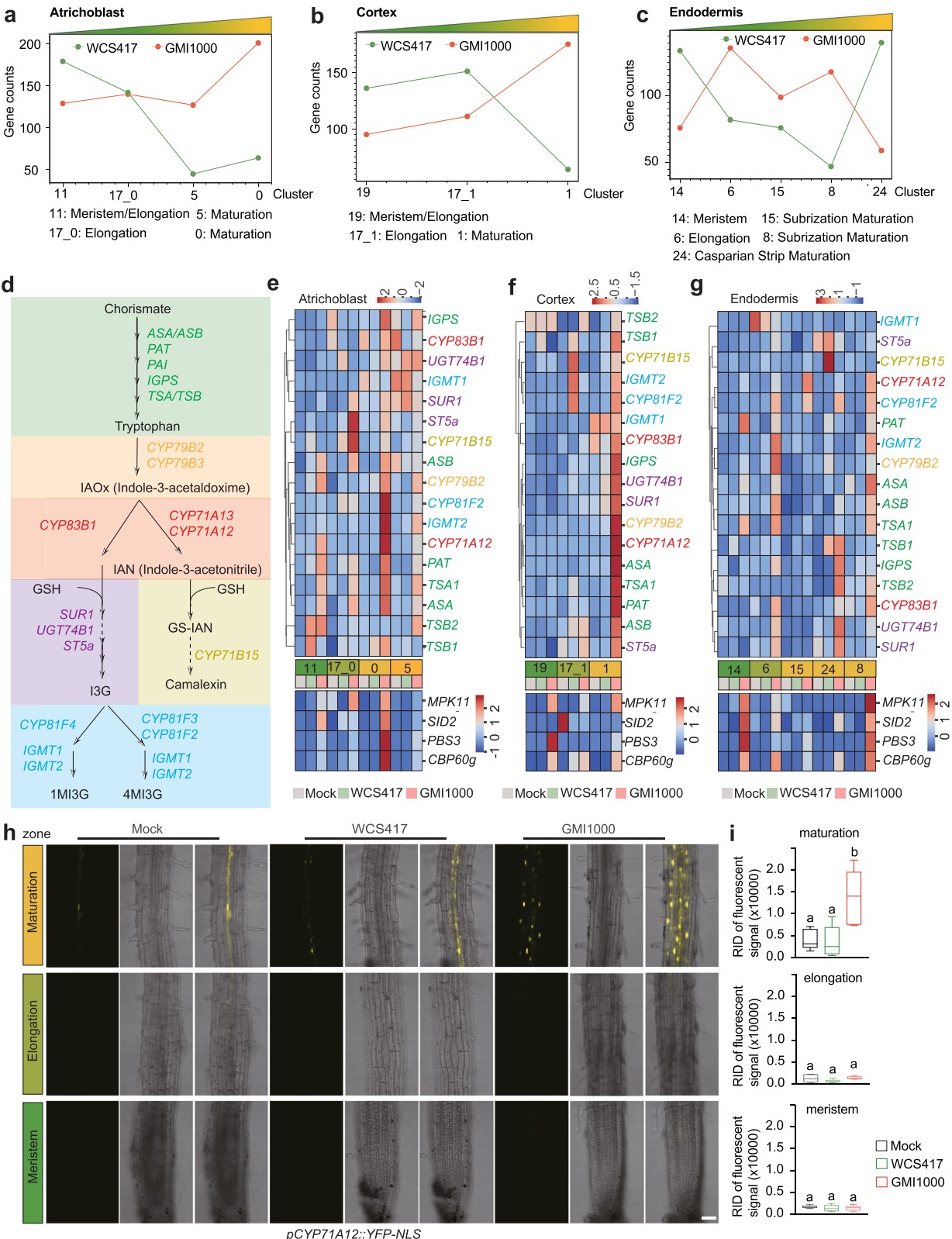

(Fig. 6c). Our genetic and microbiome profiling results strongly suggest that THAS1 is involved in sculpting microbiome structure in response to GMI1000 infection. It is noteworthy that the *thas1* mutant does not affect *Ralstonia* resistance in the mono-association system (Supplementary Fig. 11)[43], and thus the role of THAS1 in reshaping root microbiome is more likely because of its effect on microbiome structure rather than direct pathogen resistance.

To further elucidate which microbiome members are affected by the *thas1* mutant and their potential biological relevance, we compared the relative abundance of microbes from different genera. We observed that the relative abundance of ASVs from Oxalobacteraceae was significantly enriched in response to GMI1000 infection in the wild type but not in the *thas1* mutant (Fig. 6d, e). Oxalobacteraceae was recently reported to be a growth-promoting microbe in maize under

**Fig. 4 | Localized expression of phytoalexin biosynthesis genes in response to GMI1000. a–c** The line charts show the number of differentially expressed genes corresponding to the maturation, elongation, and meristem zones of the root atrichoblast, cortex, and endodermis, respectively. **d** Overview of the phytoalexin biosynthesis pathway. **e–g** Heatmap illustration of the Z-score normalized expression levels of genes from phytoalexin biosynthesis pathway and several immune or SA-related marker genes, including *PBS3, MPK11, CBP60g,* and *SID2.* **h** Validation of the maturation cell-specific induction pattern of *CYP71A12* using its promoter-driven reporter line (*pCYP71A12::YFP-NLS*), roots were inoculated with mock (10 Mm MgSO$_4$), GMI1000 or WCS417 (OD$_{600}$ = 0.05) for 6 h. Images

correspond to the meristematic zone (MZ), elongation zone (EZ), and maturation zone (MZ). The scale bar represents 50 μm. **i** Quantitative analysis of YFP signal intensities of the *CYP71A12* reporter line after different treatments and from different regions. $n$ = 6 from two independent experiments. Results are presented as mean ± standard error of the mean (SEM). Different letters indicate statistically significant differences ($P < 0.05$) between means by ANOVA and Tukey's test. Box plots show the median (horizontal bar), 25th (bottoms of boxes), and 75th (tops of boxes) quartile range (QR), and non-outlier data value (upper and lower whiskers) of each index.

nitrogen deprivation conditions[44], suggesting that microbes from Oxalobacteraceae may harbor diverse beneficial root-associated microbes. Interestingly, the relative abundance of microbes from Comamonadaceae was significantly depleted in Col-0 upon GMI1000 infection; however, it was significantly enriched in the *thas1* mutant (Fig. 6e). This indicates that wild-type plants might directly or indirectly suppress the colonization of Comamonadaceae strains during *Ralstonia* infection, while this effect is genetically dependent on *THAS1*.

To further investigate the potential biological relevance of enriched ASVs from Oxalobacteraceae and Comamonadaceae, we employed a high-throughput dilution-based microbial isolation approach to isolate root-associated microbiome members[45]. We observed that ASV9, belonging to Oxalobacteraceae, was significantly enriched in the Col-0 root microbiome after GMI1000 infection in natural soil, whereas this enrichment was nearly abolished in the *thas1* mutant (Fig. 6f). Meanwhile, the relative abundance of ASV5 was notably higher in Col-0 compared to the *thas1* mutant even without GMI1000 infection (Fig. 6f). Through this approach, we successfully isolated four strains belonging to Oxalobacteraceae (Supplementary Data 4 for 16S sequences and the corresponding ASVs from microbiome sequencing). We found that the isolated strain *Massilia* sp. 22G3 shows 96.065% sequence similarity to ASV9, and *Undibacterium* sp. 10G7 shows 96.296% sequence similarity to ASV5. Subsequently, we assessed whether these strains belonging to the root-enriched Oxalobacteraceae can confer protection against *Ralstonia* GMI1000 infection using an in-planta protection assay. In the assay, we mixed GMI1000 with each tested strain at a 1:1 ratio and inoculated the mixture onto roots of 5-day-old seedlings. Both of these two Oxalobacteraceae strains can strongly protect roots from GMI1000 infection. In contrast, we also isolated two Oxalobacteraceae strains which are not drastically changed in response to GMI1000 infection. For example, *Massilia* sp. 1H4 shows 99.767%, 99.07% and 98.605% similarities to ASV24, 55 and 107, respectively; *Pseudoduganella* sp. 9C5 shows 100%, 99.302% and 98.605% similarities to ASV55, 24 and 107. Both ASV55 and 107 have very low relative abundance in detected Oxalobacteraceae ASVs and do not show enrichment upon GMI1000 infection. Those two strains (1H4 and 9C5) also failed to protect roots from GMI1000 infection. Our results suggest that some enriched Oxalobacteraceae strains upon GMI1000 infection in wild-type root microbiome tend to protect roots from GMI1000 infection, while some low-abundance or not enriched Oxalobacteraceae strains fail to protect (Fig. 6g, h). However, since we only isolated 4 Oxalobacteraceae strains in this project, whether this family is mainly responsible for disease suppression requires more strain isolation and functional validation in the future. We further tested whether 10G7 and 22G3 outcompete the GMI1000 growth on the root surface. We found that 22G3 co-inoculation has no significant effects on GMI1000 root colonization level, and 10G7 co-inoculation weakly outcompetes GMI1000 growth. This evidence suggests that direct pathogen out-competing might not be the major reason underlying the biocontrol activities (Supplementary Fig. 12). Furthermore, we identified three strains belonging to Comamonadaceae, which was a family depleted upon *Ralstonia* GMI1000 infection (Fig. 6e). We hypothesized that these

depleted microbes might not have a protective effect, and indeed, all three tested Comamonadaceae strains failed to protect seedlings from GMI1000 infection (Fig. 6g, h). Our root microbiome composition analysis confirmed that: (1) GMI1000 infection triggers extensive root microbiome reshaping; (2) this microbiome reshaping process is genetically blocked in *thas1* mutant.

## Discussion

Understanding how eukaryotic organisms interact with beneficial microbes while concurrently restricting pathogenic microbes within complex microbiomes represents a pivotal question in the field of host-microbiome interactions. Investigating this phenomenon necessitates the characterization of distinct host responses to both beneficial and pathogenic microbes. However, plant roots, being highly heterogeneous tissues and typically growing inside the soil, present formidable challenges in systematically unraveling the mechanisms governing root-microbe interactions. In this study, we addressed this challenge by employing a mesh-based hydroponic seedling growth system in conjunction with our recently developed single-nucleus RNA-seq approach. Through this integrated approach, we comprehensively elucidated cell-type- and developmental stage-specific responses to both beneficial and pathogenic microbes in roots. The heightened cellular resolution afforded by our methodology facilitated the discernment of enriched gene expression patterns at the single-cellular levels. Subsequently, this enabled the identification and validation of pathways associated with translational regulation and triterpene biosynthesis necessary for root-microbe interactions.

Over the past decades, research in the field of plant immunity has systematically elucidated how plants perceive danger, primarily through the recognition of elicitors[46,47]. However, a more challenging question arises regarding how plants discern between beneficial and pathogenic microbes, considering that both types of microbes share similarities in conserved immune elicitors[22]. One possibility is that over several days of interaction, pathogenic microbes cause damage to roots, while beneficial microbes promote growth[11], leading to differential root responses as a result. For example, pathogenic strains like GMI1000 induce epidermal cell death within 12 h of inoculation[9], whereas beneficial strains like WCS417 promote root hair and lateral root growth after 2 days of interaction[19]. To investigate whether and how roots recognize different microbes, it is crucial to profile early-stage interactions before damage and growth-promoting effects occur. Traditional protoplast-based single-cell approaches are impractical for accurately profiling early-stage root responses within the first few hours due to the time required for protoplast preparation. However, our protoplast-free single-nucleus RNA-seq approach enabled us to accurately profile early-stage root-microbe interactions at the 6 h time point. Our data suggest that plant roots can recognize and differentially respond to different microbes in a cell-autonomous manner during the early stages of interaction. By systematically characterizing these specialized molecular responses to beneficial and pathogenic microbes, we lay the basis for further investigating the regulatory mechanisms and signals that enable roots to distinguish between "friends and foes."

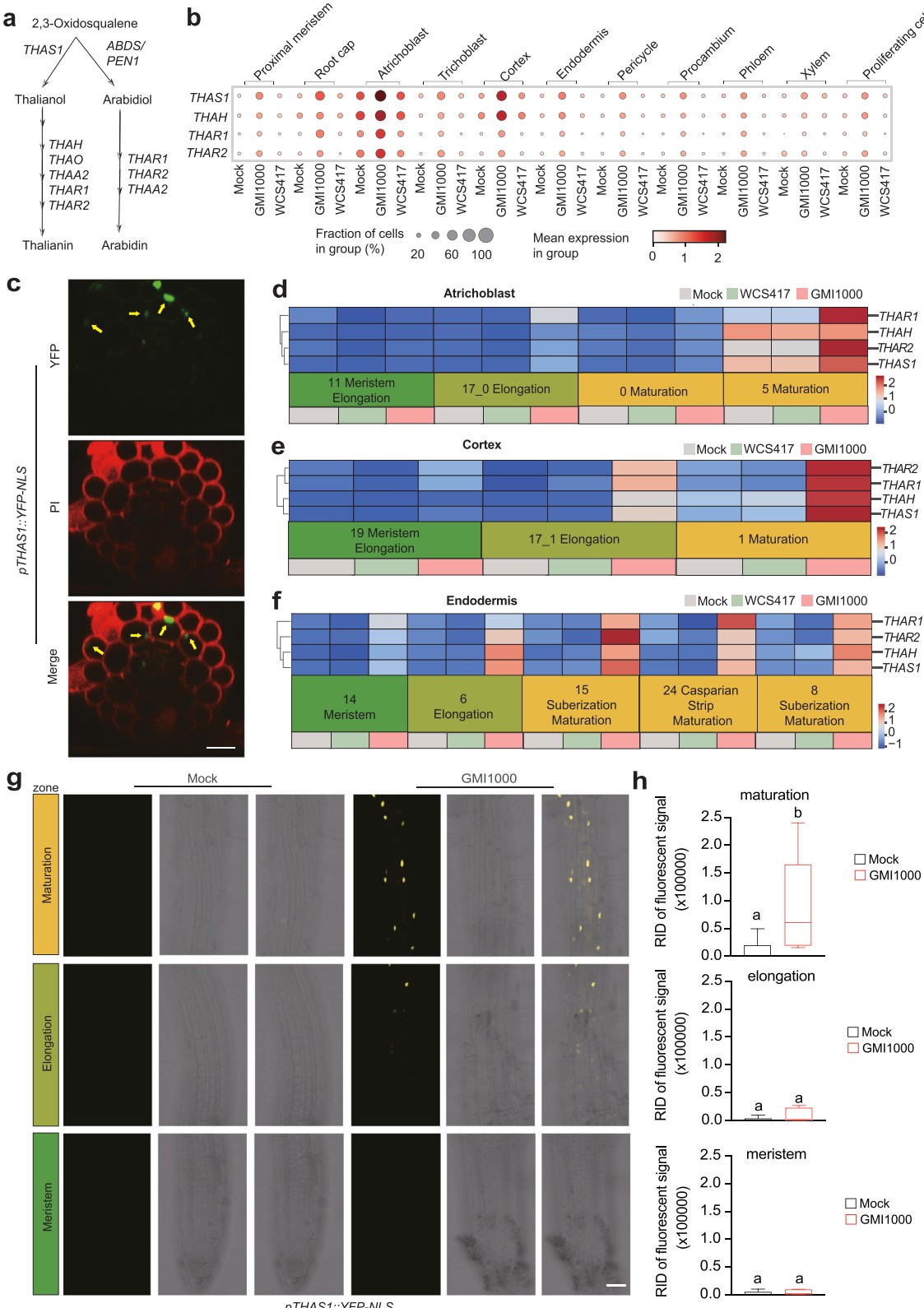

*pTHAS1::YFP-NLS*

Although it has long been proposed that plants lack specialized immune cells, our work confirms significant heterogeneity in immune responses among different root cell types. Notably, root maturation cells exhibit some degree of specialized immune responses, particularly in the cell type-specific upregulation of secondary metabolite biosynthesis genes during early-stage root-microbe interactions. While previous studies suggested that root immune responses are developmental stage-specific and that root maturation zones largely lose immune responsiveness to elicitors[9]. By examining the expression patterns of core immune-responsive genes in different sub-cell clusters with varying developmental stages[25], we found that root maturation cells exhibit even higher numbers of immune-related differentially expressed genes (DEGs) in response to *Ralstonia* pathogens compared to juvenile cells in atrichoblast and cortex cell types. Furthermore, we

**Fig. 5 | Localized expression of triterpenoid biosynthetic genes in response to GMI1000. a** A simplified pathway shows the key enzymes catabolizing the biosynthesis of two triterpenoid products (thalianin and arabidin). **b** Several triterpenoid biosynthetic genes are strongly induced by GMI1000 but not WCS417. **c** The expression pattern of *pTHAS1::YFP-NLS* in root maturation zone, Maximum projections of transverse sections are shown. Yellow arrows indicate atrichoblast cells or cortex cells. Scale bars, 20 μm. The experiment was repeated 3 times with similar results. **d–f** Z-score normalized expression heatmap illustration of the expression patterns of *THAR1*, *THAR2*, *THAH*, and *THAS1*. **g** Validation of the maturation cell-specific induction pattern of *THAS1* using its promoter-driven

reporter line (*pTHAS1::YFP-NLS*), roots were inoculated with mock (10 mM MgSO₄) or GMI1000 (OD₆₀₀ = 0.05) for 6 h. Images correspond to the meristematic zone (MZ), elongation zone (EZ), and maturation zone (MZ). The scale bar represents 50 μm. **h** Quantitative analysis of YFP signal intensities of the *THAS1* reporter line after different treatments and from different regions. $n = 6$ from two independent experiments. Results are presented as mean ± standard error of the mean (SEM). Different letters indicate statistically significant differences ($P < 0.05$) between means by ANOVA and Tukey's test. Box plots show the median (horizontal bar), 25th (bottoms of boxes), and 75th (tops of boxes) quartile range (QR), and non-outlier data value (upper and lower whiskers) of each index.

observed that genes related to tryptophan-derived antimicrobial metabolites biosynthesis pathways are predominantly induced by GMI1000 in the maturation zone, highlighting the specificity of immune responses in this region. We further validated this using a promoter-driven reporter line and bioimaging system. Although the previous study suggests that cells from the root maturation cell zone show weakened responsiveness of a few marker genes to pure elicitors[9], we found that maturation cells were still able to strongly activate antimicrobial metabolites-related immune genes in response to root pathogens. Interestingly, the expression pattern of genes related to the thalianol biosynthesis pathway is very similar to the phytoalexin biosynthesis pathway. This leads to the question of whether root maturation cells are specialized cells for the expression of all or most secondary metabolic pathways. We further examined the expression patterns of genes related to other root microbiome interaction-related secondary metabolic pathways, including genes related to scopoletin and flavonoid biosynthesis[48–50]. However, we did not see a maturation cells-specific response for those pathways (Supplementary Fig. 13). That indicates root maturation cells show specificity for the induction of phytoalexin and thalianol biosynthesis genes upon *Ralstonia* infection. This might be because *Ralstonia* prefers to infect plants through lateral root emergence sites (maturation zone)[51], and thus plants evolved the immune responsiveness in the maturation zone over long-term co-evolution to combat the expected *Ralstonia* invasion. Overall, our analysis highlights the power of the high-resolution snRNA-seq approaches for comprehensively dissecting the complex and localized immune responses in heterogeneous root tissues.

The emerging evidence suggests that two major conserved host responses to root microbes are the root endodermal barrier formation and tryptophan-derived antimicrobial metabolites biosynthesis. Our data, along with previous cell sorting-based cell type-specific RNAseq studies, indicate that rhizosphere microbes can induce the expression of suberin biosynthesis genes in root endodermis cells[28]. This trend is also observed in recent root cell type-specific translatome profiling studies, which found that the pathogenic fungus *Verticillium* suppresses the translation of root endodermal barrier genes to promote propagation[52]. Mutants defective in root endodermal barrier formation not only broadly affect root-commensal interactions[28,53], but also affect vascular fungal pathogen resistance[52]. Additionally, both our data and cell type-specific translatome profiling suggest that root pathogen infection activates the tryptophan-derived antimicrobial metabolites biosynthesis pathway[52]. The camalexin biosynthesis gene *CYP71A12* is a general non-self-responsive gene showing induction in response to diverse microbes[29]. Mutants in the tryptophan-derived antimicrobial biosynthesis pathway dampen the compatible interactions between roots and rhizosphere fungal commensals or beneficial *Pseudomonas*[54,55] and also decrease root resistance to vascular fungal pathogen *Verticillium*[52]. Together, our work and other studies suggest that different root vascular pathogens, such as *Ralstonia* and *Verticillium*, share similarities in root immune responses, and root endodermal barrier and antimicrobial compounds play broad roles in regulating root-microbe interactions.

Plant secondary metabolites play a crucial role in shaping or sculpting microbiome structure in certain conditions[56]. Triterpenes have been reported to broadly regulate the growth of diverse rhizosphere microbes[42]. However, the biological context in which this pathway functions to sculpt the microbiome remains elusive. Based on high-resolution cell type-specific gene function analyses, we identified diverse triterpene biosynthetic genes that show cell-type-enriched induction in response to pathogens, especially in the outer cell types such as cortex and atrichoblast. We further confirmed that mutants defective in triterpene biosynthesis can largely block the ability to reshape the root microbiome in response to *Ralstonia* infection. This furthers our understanding of the biological relevance of the known triterpene biosynthesis pathway, which is to reshape the microbiome during root pathogen infection. However, it remains unclear whether the outer cell type-enriched expression of triterpene biosynthesis genes contributes to its function during root-microbiome interactions. Future studies using promoter replacement approaches to disrupt the expression of the triterpene biosynthesis pathway would provide more information about whether cell type-specific expression patterns affect the function of triterpenes.

Given the considerable interest in root immune responses and the intricate interactions between roots and both beneficial and pathogenic microbes, we have established a website (https://zhailab.bio.sustech.edu.cn/sn_microbe) for the scientific community to delve deeper into gene expression patterns at a single-cell resolution. Leveraging this dataset, we have already uncovered mechanistic insights into root-microbe interactions. Our dataset will provide critical clues for further identifying plant genes implicated in root-microbe interactions and influencing the composition of root-associated microbiomes.

## Methods

### Plant material and growth condition

*Arabidopsis* seeds were surface sterilized with 20% bleach for 1–2 min, followed by 75% ethanol for another 1–2 min, and then washed three times with sterilized water. Afterward, seeds were imbibed at 4 °C in the dark for 4 days. Seedlings were grown in a growth chamber (Fujian JIUPO, BPC500 H, at 23 °C under a 16 h light/8 h dark period with light intensity at 100 μmol m$^{-2}$ s$^{-1}$) for 5–7 days. Seedlings were transferred to peat pellets or natural soil after that, and plants in soil were grown at 23 °C growth room under a 10 h light/14 h dark period with light intensity at 100 μmol m$^{-2}$ s$^{-1}$. For the 48-well plate assay, plants were grown in 48-well plates with liquid 1×Murashige and Skoog (MS) medium containing 2% (w/v) sucrose for 10 days (16 h Light/8 h Dark) and transferred to fresh liquid 1 × MS medium without sucrose for 2 more days before inoculation. T-DNA insertion lines were obtained from the Arabidopsis Biological Resource Center (ABRC) for the following genes: AT3G53020 (SALK_045401), AT1G22780 (SALK_043342), AT1G26630 (SALK_003799), AT5G19510 (SALK_067603), AT1G30230 (SALK_102754), AT5G02870 (SALK_029203), AT5G39740 (SALK_110811), AT5G41520 (SALK_119578), AT2G42740 (SALK_206632), AT1G66580 (SALK_126907), AT5G60390 (SALK_035228), AT5G45775 (SAIL_1231_C04), and AT5G60670 (SALK_124523).

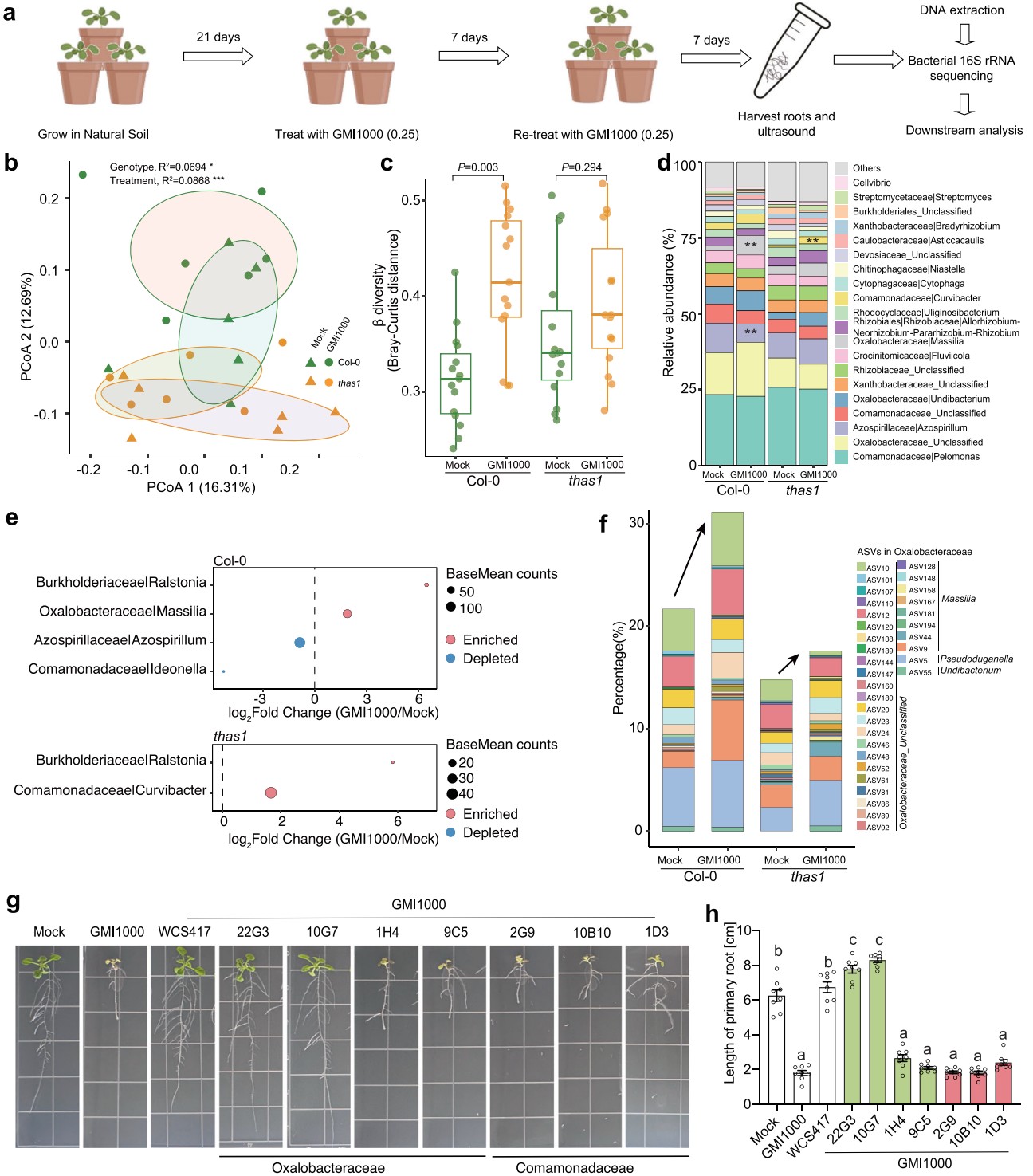

**Fig. 6 | THAS1 is required for reshaping the root microbiome in response to GMI1000 infection in natural soil. a** Schematic diagram showing the design and sample preparation for microbiome sequencing study, some elements of this figure are Created in BioRender. Qiuhua, Y. (2025) https://BioRender.com/l80x360. **b** Principal coordinates analysis based on Bray–Curtis distance was performed for all samples from different genotypes and treatments. (PERMANOVA by using vegan adonis2, $n = 6$ replicates for each group). **c** Differences in $\beta$ diversity among treatments were estimated based on the Bray–Curtis distance matrix of all samples within genotypes. Box plots show the median (horizontal bar), 25th (bottoms of boxes), and 75th (tops of boxes) quartile range (QR), as well as non-outlier data value (upper and lower whiskers). Each dot represents the pairwise distance between samples within a group ($n = 15$ each paired comparison). *P*-values were calculated using the Student's *T*-test (two sides). The *P*-values are labeled in the

figure. **d** The relative abundance of the top 20 abundant genera in the roots of different genotypes. ** represents the adjusted *P*-value below 0.01, calculated using DESeq2 with default parameters. **e** Differential abundant genera were identified by using Deseq2. The sizes of each dot represent the average counts of each genus in all samples. Different colors indicate enriched or depleted genera. **f** The relative abundance of ASVs within Oxalobacteraceae. **g** The protecting effects against GMI1000 infection in isolated strains from Oxalobacteraceae and Comamonadaceae. The images were taken 10 days after inoculated with different microbes. **h** Quantification of the primary root length from (**g**) ($n = 8$ samples). The experiment was repeated twice with consistent phenotypes. Results are presented as mean ± standard error of the mean (SEM). Different letters indicate statistically significant differences ($P < 0.05$) by ANOVA and Tukey's test.

## RNA extraction and qRT-PCR

For qRT-PCR assay, total RNA was extracted with the FastPure Plant total RNA isolation Kit (Vazyme 7E602I2) and quantified by nanodrop. Approximately 1 µg of RNA was reverse-transcribed to cDNA with HiScript III All-in-one RT SuperMix (Vazyme Q711-03). The qPCR was performed with ChamQ Universal SYBR qPCR Master Mix (Vazyme R333-01) with the housekeeping gene EIF4A as an internal control. The primer pairs used for qRT-PCR are listed in Supplementary Table 3.

## Confocal-based bio-imaging analysis

The fluorescence imaging in this work was performed with an LSM880 confocal microscope (Zeiss). For the imaging experiment, approximately 10 sterilized and stratified seeds of the *pTHAS1::YFP-NLS* (Col-0 background) transgenic lines were sown on a 1/2×MS with 1% sucrose solid plate and placed under long-day conditions. After 4–5 days, pictures were taken under a 40× water immersion objective. For cell type-specific expression pattern analysis, we used a Z-scan to dissect YFP signals in different cell layers. The excitation and detection windows were listed below: YFP 514 nm and the 518–617 nm; PI 561 nm, 616–718 nm.

For the bio-imaging of *pCYP71A12::YFP-NLS* (in Dex: AvrRpm1 inducible line background[57]) and *pTHAS1::YFP-NLS*, seeds were sown on a 1/2 × MS with 1% sucrose solid plate and placed in a long-day condition. Seven to eight days old seedlings were transferred to liquid 1/2 × MS media without sucrose with GMI1000 or WCS417 (OD$_{600}$ = 0.05), and liquid 1/2 × MS medium supplemented with an equal volume of 10 mM MgSO$_4$ was used as a mock control. For the *pAT3G30600::YFP-NLS* transgenic line, 10-day-old seedlings were transferred to liquid 1/2 × MS media without sucrose. After treatment for 6 h, the YFP signal was observed under a 20× objective. YFP was excited at 514 nm.

For quantifying the nuclear-localized fluorescence intensity of the *CYP71A12* reporter and *THAS1* reporter, confocal images were analyzed using Fiji software.

## Treatment and sampling of snRNA-seq samples

For snRNA-seq and qRT-PCR experiments, *Arabidopsis* seedlings were grown in 48-well plates with liquid 1 × MS medium containing 2% (w/v) sucrose for 10 days, and then the media was changed to fresh liquid MS medium without sucrose for 2 more days. For bacterial inoculation, *P. fluorescens* WCS417 and *Ralstonia solanacearum* GMI1000 were washed twice and resuspended in 10 mM MgSO$_4$ to remove potential exudated elicitors during overnight culture. WCS417 and GMI1000 were diluted to OD$_{600}$ = 0.5 in 10 mM MgSO$_4$ and then added to the liquid rhizosphere to a final OD$_{600}$ of 0.05. Liquid MS medium supplemented with an equal volume of 10 mM MgSO$_4$ was used as a mock control. For qRT-PCR the roots were harvested at 2 h, 6 h, 24 h, 30 h, and 48 h after inoculation, and about 5–10 roots were pooled as one sample. For snRNA-seq, 12-day-old whole roots (5–7 cm) were pooled from 30–40 seedlings at 6 h after inoculation as one sample. After harvest, the roots were snap-frozen in liquid nitrogen, and stored at −80 °C.

## Nuclei isolation and 10× snRNA-seq library construction

The roots (5–7 cm whole roots) were chopped thoroughly in ice-cold nuclei isolation buffer [1× NIB buffer (Sigma-Aldrich, #CELLYTPN1), supplemented with 1 mM dithiothreitol (DTT, Thermo Fisher, R0861), 1× protease inhibitor (Sigma, 4693132001) and 0.4 U/µL murine RNase inhibitor (Vazyme, R301-03)]. Then the lysate was passed through a 40 µm strainer (Sigma, CLS431750-50EA) and centrifuged at 500 × *g* for 5 min at 4 °C. The pellet was gently resuspended in 500 µL of nuclei isolation buffer and 0.5 µL of 1 mg/mL DAPI was added. To remove the remaining tissue debris, the nuclei suspension was filtered through a 20 µm cell strainer (pluriStrainer Mini, 43-10020). For cell sorting, the nuclei were loaded to a flow cytometer (Sony MA900) equipped with a

100 µm nozzle. A total of 100,000 nuclei for each sample were sorted into a 15 mL tube with 2 mL collection buffer (1× PBS, supplemented with 1% BSA and 0.4 U/µL murine RNase inhibitor). The nuclei were centrifuged at 500 × *g* for 5 min at 4 °C and resuspended in 50 µL collection buffer. The quality and yield of nuclei were checked under a fluorescence microscope using the DAPI channel. About 16,000 nuclei of each sample were loaded into the 10× Genomics chip for cell barcoding. The libraries were prepared following the manufacturers' instructions with a 10 × Chromium Single Cell 3′ Solution v3.1 kit. The pair-end libraries were sequenced with the Illumina Nova-seq instrument.

## Data processing and cell type annotation for snRNA-seq analysis

The raw reads were preprocessed by Cell Ranger (v6.0.0)[58] and aligned to the *Arabidopsis* TAIR 10 genome, generating the h5 files for each sample. Then we used SCANPY package (v1.8.0)[59] in Python to read the files and used ScDblFinder (v1.10.0)[60] to remove the doublet. After that, we kept only the cells with gene counts larger than 300 and less than 3500, and UMI counts larger than 500 and less than 6000. In addition, cells with more than 0.05% transcripts from mitochondria or chloroplasts were removed.

We then adopted Seurat (v. 4.3.0) to integrate the six matrixes following the official guidance (https://satijalab.org/seurat/articles/integration_introduction). We used SCTransform to normalize and call the top 3000 highly variable features for each matrix and selected the shared high variable features for canonical correlation analysis (CCA) to identify the integrating anchors. After that, we used the 'scanpy.pp.neighbors' function to call the nearest-neighbor graph (default parameter) and performed the Leiden algorithm (scanpy.tl.leiden, resolution = 0.6) on the graph for clustering. The Uniform Manifold Approximation and Projection (UMAP) algorithm was used for visualization (scanpy.tl.umap, resolution = 0.2). For annotation, we first used CELLEX (CELL-type Expression-specificity, v1.2.2) to call the cluster-enriched genes from our data and the reference data, and annotated the clusters according to their Intersection of Union level (Supplementary Fig. 2c, d).

## Differential expression analysis and GO enrichment analysis

We adopted Diffxpy (https://github.com/theislab/diffxpy) to identify the Differential Expressed Genes (DEGs) from the treatment of GMI1000 or WCS417 against the Mock sample within each cluster. Genes expressed in less than three cells in the two comparing clusters would be masked for DEG calling. From the Diffxpy calculation result, genes with *P*-value < 0.05 (Wald test) and log2 | (fold change) | > 1 would be considered significantly different from Mock. We used the enricher function in R package clusterProfiler (v4.6.0)[61] for GO enrichment analysis of the DEGs, and the dotplot illustration of GO results was conducted by the ImageGP website[62].

## Plasmid construction and plant transformation

For *pTHAS1::YFP-NLS*, a promoter fragment (−2000 bp upstream of the translation starting site) was PCR amplified. For *pCYP71A12::YFP-NLS*, a genomic fragment (−2472 bp upstream of the translation starting site) of the *CYP71A12* locus was PCR amplified[63]. For *pAT3G30600::YFP-NLS*, a 2000-bp promoter fragment upstream of ribosome genes. The fragments were assembled into EcoRI/NcoI-linearized *pCAMBIA1300* binary vector followed by a YFP and nucleus localization sequence insertions to generate the *pTHAS1::YFP-NLS* plasmid, *pCYP71A12::YFP-NLS*, and *pAT3G30600::YFP-NLS* through Gibson assembly, respectively. *pCYP71A12::YFP-NLS* transgenic line was constructed in a DEX-avrRPM1 inducible line background[57] for other purposes, however, we did not use any dexamethasone treatment in our mock and treatment groups.

The binary constructs were transformed into *Arabidopsis* by using *Agrobacterium tumefaciens*-mediated transformation (strain GV3101),

followed by selection on hygromycin (50 μm)-containing medium. The primer pairs used for PCR are listed in Supplementary Table 3.

### Natural soil growth substrates

The natural soil was collected from Xishuangbanna Tropical Botanic Garden (E101°27′, N21°92′) and Yuanjiang Savanna Ecosystem Research Station (E102°10′, N23°28′). Natural soil was sieved through a 2 mm sieve. Soil from different locations was 1:1 mixed as a natural soil mixture. Then, a mixture substrate composed of equal volumes (1:2:2:1) of mixed natural soil, vermiculite, clay, and perlite as soil substrates for natural soil experiments. The soil was scooped into $6 \times 6$ cm pots for plant transplantation. Vermiculite, clay, and perlite were autoclaved at 121 °C for 20 min before use.

Col-0 and *thas1* mutants were grown on $1 \times$ MS medium containing 2% sucrose and 1.2% agar for 7 days. Seedlings were grown in a growth chamber (Fujian JIUPO, BPC500 H, at 23 °C under a 16 h light/ 8 h dark period with light intensity at 100 μmol m$^{-2}$ s$^{-1}$). The seedlings were then transferred to natural soil and allowed to grow for 21 days before irrigation with GMI1000 (OD$_{600}$ = 0.25 in water). The seedlings were irrigated with GMI1000 again 7 days after the first irrigation. Eventually, 7 days after the second time GMI1000 irrigation, the plants were removed from the pot. For root sampling, the roots of each plant were cleaned and sonicated for 5 seconds to remove the loosely attached soil. After drying the surface water, the root samples were quickly frozen in liquid nitrogen as microbiome samples.

### DNA extraction and microbiome sequencing

For DNA extraction, DNA was extracted using the Power Soil DNA Isolation Kit (Qiagen, Germany) following the manufacturer's protocol. The DNA samples with a concentration higher than 20 ng/μLwere used for microbiome sequencing. The DADA2[64] pipeline was used to generate amplicon sequence variants (ASVs), and taxonomic classification was performed using the SILVA database (release 138) with pre-trained naive Bayes classifier based on the representative sequence of each ASV[65,66]. The V3-V4 region of the 16S rRNA gene was amplified using primers 349F 5′-ACTCCTACGG-GAGGCAGCA-3′ and 806R 5′-GGACTACHVGGGTWTCTAAT-3′ for amplicon sequencing.

### *Ralstonia* infection protecting assay for rhizosphere isolates

Sterilized wild-type seeds (Col-0) were grown on $1/2 \times$ MS with 1% sucrose solid plate (1.2% agar) and were placed in long-day condition. After 5 days of growing, 4 healthy seedlings were transferred to $1/2 \times$ MS (without sucrose) solid plate (1.2% agar) for root surface inoculation with mixed microbes. A mixture of two microbes (GMI1000 and each strain) was 1:1 mixed. Ten μL mixture inoculum were inoculated onto roots at a final concentration of OD$_{600}$ = 0.001. The phenotype was taken 10 days after inoculation.

For the CFUs calculating, We utilized antibiotic resistance to distinguish the GMI1000 and 22G3/10G7. We transferred the GMI1000 strain with a gentamicin resistance vector (pBB-GFP), while 22G3/10G7 has no gentamicin resistance. Four roots were mixed as a sample to measure CFUs after inoculation for 7 days.

### *Ralstonia* resistance assay on plate system[43]

Sterilized seeds of Col-0 and *thas1* were grown on $1/2 \times$ MS with 1% sucrose solid plate (1.2% agar) and then were placed in long-day condition. After 5 days of growing, 8 healthy seedlings were transferred to $1 \times$ MS (without sucrose) solid plate (1.2% agar) containing 0.5% MES for root surface inoculation with GMI1000 (5 μL) inoculum was applied onto roots at a final concentration of OD$_{600}$ = 0.0001. 4 shoots were taken as a sample at 3 days post-inoculation. The bacterial solution was spread on plates, and the number of colonies was counted after 2 days of inoculation at 28 degrees.

### DNA extraction and bacterial 16S rRNA gene sequencing

The DNA of all root samples was extracted using a Power Soil DNA Isolation Kit (Qiagen, Germany). DNA samples with a concentration of more than 20 ng/μL were used for 16S rRNA gene sequencing. We amplified the V3-V4 regions of the 16S rRNA gene using the primers 349F (5′-ACTCCTACGGGAGGCAGCA-3′) and 806R (5′-GGACTACHV GGGTWTCTAAT-3′). The library was prepared using NEBNext® Ultra™ II DNA Library Prep Kit (New England Biolabs, USA), and library concentrations were determined using a Qubit 4.0 fluorometer. Paired-end (250 bp) amplicon sequencing was performed on an Illumina NovaSeq 6000.

### Microbiome data analysis

Raw sequencing reads of amplicon sequencing were first filtered using fastp v.0.14.1[67]. Adapter sequences and primers were removed using Cutadapt v.4.0[68]. Retained reads were further filtered and taxonomically analyzed using QIIME2 v.2022.2[69]. In brief, DADA2 was used to filter and denoise sequences, remove chimaeras, identify representative sequences, and generate a unique amplicon sequence variants (ASVs) table[64]. ASVs were taxonomically annotated via a pre-trained naive Bayes classifier on the basis of the SILVA database (release 138)[65,66]. Sequences annotated as chloroplasts and mitochondria were considered host contamination and were thus removed. ASVs that were present in more than 3 samples were used for downstream analysis.

### Beneficial WCS417-mediated lateral root growth-promoting assay

Col-0, *cdc123*, *hem1* and ribosome-related mutants seeds were sown on a $1/2 \times$ MS (no sucrose) solid plate (0.86% agar) under mock and WCS417 inoculation conditions. For WCS417 plates, the medium was added with WCS417 (final concentration OD$_{600}$ = 0.00001) before medium solidification (at around 50 degrees). All plants were placed under long-day (16 h light) conditions. The lateral root phenotype was taken 11 days after germinating. The shoot fresh weight was measured at 21 days after germinating and 4 roots were taken as a sample to measure CFUs.

### Reporting summary

Further information on research design is available in the Nature Portfolio Reporting Summary linked to this article.

## Data availability

The snRNA-seq and microbiome data were deposited in the China National Center for Bioinformation database under PRJCA019084 and PRJCA026420, respectively. Source data are provided with this paper.

## Code availability

All the codes for snRNA-seq analysis in this work can be found at Zhai lab's GitHub site: https://github.com/ZhaiLab-SUSTech/snRNA-seq_microbes.

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

## Acknowledgements

We thank Dr. Cara Haney and Dr. David Thoms for critical reading of the manuscript. We thank Drs. Guoyong Xu for kindly gifting dst5 and dst7 seeds, Hongwei Guo for gifting plasmids, and Yuping Qiu for the help with confocal imaging. The project is supported by the Biological Breeding-National Science and Technology Major Project (2023ZD04073), National Natural Science Foundation of China (32270286, 32325031); Y.S. was supported by the Stable Support Plan Program of Shenzhen Natural Science Fund Grant (20220815160107001), and the Shenzhen Science and Technology Program (Grant No. RCYX20231211090408015 and ZDSYS20230626091659010); and Guangdong Innovative and Entrepreneurial Research Team Program 2023ZT10Y013. Y.L. is supported by the Natural Science Foundation of the Higher Education Institutions of Guangdong Province (2022KTSCX115) and Guangdong Basic and Applied Basic Research Foundation (2023A1515011997).

## Author contributions

Y.S., Y.L., J.Z., and A.H. designed the project. Q.Y., Y.L., K.G., and Y.S. performed the experiments. Z.W.L., K.G., Z.W., Q.Y., and Z.J.L. analyzed the data. Y.H. and X.T. supported the genetic materials. Y.S., Q.Y., Z.W.L., and Y.L. wrote the manuscript.

## Competing interests

The authors declare no competing interests.
