## [Transparent Peer Review file · Nature Communications]

Comparative single-nucleus RNA-seq analysis revealed localized and cell type-specific pathways governing root-microbiome interactions

Corresponding Author: Dr Yi Song

Version 0:

Reviewer comments:

Reviewer #1

(Remarks to the Author)

In this study, Yang, Li, and Guan used single-nucleus RNA-seq to investigate the responses of Arabidopsis roots colonized by either beneficial (*Pseudomonas simiae* WCS417) or pathogenic (*Ralstonia solanacearum* GMI1000) bacteria. Their data indicate that roots show both common and distinct responses to beneficial and pathogenic microbes. They found that genes related to ribosome complex formation, translation, are specifically induced by WCS417 in the proximal meristem. They also showed that GMI1000 can induce immune genes (especially glucosinolate-related genes) in mature epidermis, while WCS417 did not. The authors identified an induction of triterpene biosynthesis genes not observed in a previous bulk RNA-seq study and showed that one of these genes is required for root microbiome assembly, highlighting the utility of single cell analysis. Overall, this study provides new insights into cell type-specific responses of roots to colonization by bacteria with different lifestyles, with validation and some functional analysis.

The manuscript is clearly written, but some claims related to Fig. 6 require more attention. In Fig. 6, the authors explore the role of THAS1, which is induced in a cell type-specific manner upon GMI1000 infection. While this section suggests that THAS1 is involved in microbiome assembly and responses to GMI1000 infection, the data do not fully support the authors' conclusions. The authors seem to draw conclusions beyond the available data. Detailed comments are provided below.

==Major comments==

COMMENT-1

The manuscript lacks sufficient detail regarding the codes used for analyses and figure generation. To ensure reproducibility, it is important that all raw data and codes needed to replicate key analyses are made readily available. This includes detailed information about the analysis environments, such as the specific packages and versions used. Ideally, this should be provided in a containerized format using tools such as Docker. In addition, repositories for the code need more comprehensive documentation to effectively support the research community. Finally, the methods section of the manuscript should be expanded to provide more detail.

COMMENT-2

Fig3c-e is interesting. Can the authors check the colonization of bacteria in the mutants? Is the lack of growth suppression related to the impaired colonization or something downstream?

COMMENT-3

Could the authors confirm whether DST5 and DST7 are expressed in proximal meristem? This information is important to link a cell type to function.

COMMENT-4

Validation of ribosome related genes in space would be needed as this is a major finding of this study.

COMMENT-5

In Fig 5g, the authors attempted to show atrichoblast/cortex-specific expression of THAS1 using a reporter line. However,

the presented images are unclear and it is difficult to assess the cell type specificity. Please provide clearer data. Typically, a nuclear localization signal (NLS) reporter line provides a clearer signal of cell specificity, as the authors have done in Fig. 4h. The authors should also evaluate the inducibility of THAS1 by GMI1000 infection using the reporter line.

COMMENT-6

Fig 6e is confusing. It's hard to see what's being compared in each plot. A more detailed legend would be helpful. The fold change values for Col-0 seem to be incorrect. Shouldn't the authors expect a positive value for Oxalobacteraceae, since this strain is enriched upon GMI infection? Also, the x-axis says GMI100, not GMI1000.

COMMENT-7

If THAS1 recruits beneficial strains upon GMI infection, then plant growth will be reduced in the *thas1* mutant upon GMI infection in the presence of the microbiome compared to WT. I suggest that the authors test this. Comparing WT and *thas1* with GMI alone might also be informative as this will tell us if there is a microbiome independent role of THAS1 in defense against GMI.

COMMENT-8

The conclusions drawn from Fig 6f-h are not convincing. It is not very surprising that some commensals can protect plants from GMI1000. It will take a more comprehensive screening of commensals to convincingly say that commensals recruited by plants using the THAS1 pathway protect plants.

COMMENT-9

In Line 376, the authors claimed "This indicates that wild-type plants repel the colonization of Comamonadaceae strains during *Ralstonia* infection, and this effect is genetically dependent on THAS1." Is there any evidence where plants "actively" repel Comamonadaceae during GMI1000 infection? It is possible that GMI1000 directly affects Comamonadaceae?

COMMENT-10

Appropriate statistical tests or replicate information are missing in some figures, such as Fig 3de and Extended Data Fig 1.

==Minor comments==

COMMENT-11

Showing the number of cells per merged cluster (11 clusters) for each condition (mock, colonization) would be helpful.

COMMENT-12

Fig 2c shows strong induction of auxin related genes in atrichoblast and cortex upon GMI1000 infection. Since the signal is very strong, I'd appreciate more discussion on the strain and cell type specific induction of this process.

COMMENT-13

In Extended data Fig 3c, does the number of DEGs in each cell type after different treatments reflect the number of cells analyzed or the quality of the data, such as genes detected per cell (and thus statistical power)? This needs to be assessed to make a claim. "indicating that root proximal meristem cells display the most distinct responses to beneficial and pathogenic microbes. (Line 232)"

COMMENT-14

In Fig4, they used 970 core immune genes previously identified using seedling experiment. It is not specifically root immune genes, so focusing on these genes may bias the analysis. I suggest performing more comprehensive analysis to identify developmental stage specific DEGs followed by GO enrichment or overlap analysis against the core immune genes. This will be a similar analysis to Extended Data Fig 5 but with all genes.

COMMENT-15

How do the IGS related genes look like in other cell types? Since the finding is interesting it would be informative to show the comprehensive picture of the expression of these genes in the atlas.

COMMENT-16

Fig 5b is unclear. It does not help showing THAS1 induction in GMI1000 treated roots.

(Remarks on code availability)

Reviewer #2

(Remarks to the Author)

This manuscript reports an interesting investigation of plant responses to both beneficial and pathogenic bacteria using single-nucleus RNA-seq. While the localized immune response in plant roots has been studied using transgenic reporters, our understanding of genome-wide transcriptional changes related to the spatial organization of immunity during bacterial colonization at single-cell resolution remains limited. This paper provides insight into the localized and cell-type specific pathways in plant roots in response to rhizosphere microbes. Although I am generally positive about the overall premise of the paper, there are several aspects regarding data analysis and experimental details that need to be clarified. I have

outlined my comments below.

Major comments:

- 1) How is the expression of HEM1 and CDC123 in different cell types after different treatments? Do these two genes belong to cluster-D (1055)? Do pathogen treated cells show higher expression than the beneficial bacteria? Please provide data regarding the expression of HEM1 and CDC123 in different cell types after different treatments.
- 2) How many core DEGs are overlapped in cluster 11, 17_0, 19, 17_1? Compare the expression level of the overlapped genes in these clusters.
- 3) Lines 280-281, the meristem shows the largest response to WCS417, while maturation zone shows the most significant response for GMI1000. This might be due to the different interaction/infection strategies of two bacteria, as *Ralstonia* prefers to infect plants through wounds and lateral root emergence sites. Therefore, the different immune strategies that root employed at the very early stage of the interaction may result from the colonization/infection preference of various bacteria.
- 4) Lines 282-288, It is true that maturation/differentiation zone (atrachoblast cells) loses responsiveness to pure elicitor treatment, but it shows strong response when coupled with damage (Zhou et al., 2020). Thus, maturation/differentiation zone mounts an immune response to pathogen infection due to coincidence of MAMPs and damage. The section between lines 282-288 gives the impression that the immune response of the maturation/differentiation zone was previously unknown. Please edit this section.
- 5) Elongation zone doesn't show expression in figure 4 h-l, but lines 310-311 mention strong expression. Please ensure consistency between the figure and the text.
- 6) It is interesting to see highly specific induction of CYP71A12 and THAS1 genes in the maturation zone. Do other defense related secondary metabolite genes (both snRNA-seq data and transgenic lines) show this pattern? How about the general immune related marker genes, e.g. FRK1, PER5, MPK11, WRKY33, SA related genes etc.? Do they show the same or different pattern?
- 7) THAS1-YFP localization is interesting. It seems to co-localize with cell wall/plasma membrane or outside of cells. Please include a colocalization assay to confirm THAS1-YFP is on the cell wall/plasma membrane of atrichoblast and cortex cells but not outside of cells. A transcriptional reporter (pTHAS1-YFP) might also directly demonstrate its expression location.
- 8) Lines 395-400, what about GMI1000 infection and growth in this process? It would be interesting to investigate whether GMI1000 can still infect plants or if 22G3/10G7 has the ability to kill or outcompete *Ralstonia*. Please check and compare the population levels of GMI1000 and 22G3/10G7.
- 9) Lines 387-397, "three strains (9C5, 1H4, 22G3) exhibited high 16S rRNA similarity to ASV9 (Supplementary Table 6)". "... both 22G3 (ASV9-like) and 10G7 (ASV5-like) strains". These descriptions are not consistent with Supplemental table 6, which shows 9C5, 1H4, 22G3 are ASV12 or ASV20-like. Please check and provide a justification.
- 10) Lines 438-439, GMI1000 induces cell death at 12 hpi but not at 6 hpi (Zhou et al., 2020). The data in this manuscript was obtained at 6 hpi. The bacterial ODs were different: 0.1 in Zhou et al., 2020 and 0.05 in this MS. Does the single-cell data in this manuscript show cell death related pathways at 6 hpi? Or the authors might experimentally check root cell death at 6 hpi. I wonder if the immune response triggered at 6 hpi is cell death/damage dependent or not?

Minor comments:

- 1) Line 106, delete "(GMI1000 hereafter)"
- 2) Lines 179-180, rephrase this sentence "we also observed induction of two PSKR1 ligand genes, Phytosulfokine (PSK) 1 but not PSK2,". It's confusing if PSK2 is induced or not.
- 3) Line 563, correct the sentence "and the 517-617 nm"
- 4) Line 823 and 827, duplicate reference
- 5) Line 831 and 833, duplicate reference
- 6) Figure 6e, "***" in the legend does not appear in figure. Please provide statistical method here.
- 7) Extended data Figure 1 legend is incorrect. Please edit.

(Remarks on code availability)

Version 1:

Reviewer comments:

Reviewer #1

(Remarks to the Author)

I appreciate the authors' efforts in addressing my previous comments. While many of my concerns have been resolved, several major issues remain. A key shortcoming of the manuscript, in my opinion, is limited compelling evidence that links cell type-specific gene expression changes to root-bacteria interactions. This gap significantly limits the impact of the study, as it does not convincingly demonstrate that the snRNA-seq analysis has led to new mechanistic insights, as implied by the title and abstract. Below are my detailed comments.

Revision Comment 1

As both initially and currently written, the manuscript implies that HEM1 and CDC123 are specific to the proximal meristem, and their specific expression may play an important role in WCS417-mediated plant growth promotion, which would have been an interesting finding. However, in response to my previous Comment 3, the authors have shown that CDC123 and HEM1 do not exhibit cell type specificity. This result has led me to revise my initial impression and reassess the study. A significant weakness of the manuscript is the lack of data supporting the role of cell type-specific gene induction in the

beneficial effects of WCS417. As it stands, the manuscript feels like a combination of an snRNA-seq dataset and unrelated mutant analyses. Although the authors demonstrated that Thas1 expression is cell type-specific, this gene is already known to contribute to plant-microbe interactions, limiting the impact of the finding. I recommend focusing on genes that display cell type-specific expression changes in response to WCS417 and further investigating their roles in promoting beneficial plant-microbe interactions.

Revision Comment 2

Lu et al. (2018) reported that DR5 expression is induced in the vasculature upon GMI1000 infection, but not in atrichoblast or cortex cells, which seems to contradict the results of the snRNA-seq analysis presented here. How do the authors reconcile this inconsistency? This also raises concerns about the extent to which the snRNA-seq data has contributed to our understanding of plant cell type-specific responses that govern root-microbiome interactions.

Revision Comment 3

Extended data Fig. 1 is still missing statistical tests to claim significant gene expression changes.

Reviewer #2

(Remarks to the Author)

Authors addressed all my comments but one. Please address this concern.

1) Both CDC123 (AT4G05440) and HEM1 (AT2G35110) do not exhibit specific induction in proximal meristem cells. Additionally, their induced expression appears to be nonspecific to WCS417, as high expression levels are observed in root cap, cortex, endodermis, procambium, pericycle, phloem and xylem etc. in response to GMI1000. Since 1,055 genes in Cluster-D are identified based on their highest induction in proximal meristem cells only after WCS417 but not GMI1000 treatment, this suggests that CDC123 (AT4G05440) and HEM1 (AT2G35110) are not part of 1055 genes. Consequently, there is a disconnect between the session “beneficial microbe specifically induces the expression of translation-related genes in the proximal meristem cells” and the experimental validation of “two translation regulators (CDC123, HEM1)”. It seems that these two translation regulators are not relevant to the snRNA-seq data.

Version 2:

Reviewer comments:

Reviewer #1

(Remarks to the Author)

I thank the authors for sufficiently addressing my concerns. I have no further major issues. However, I suggest that the T-DNA lines used in the revision be clearly described in the Methods section.

Reviewer #2

(Remarks to the Author)

Authors addressed my comment.

Brief introduction to major improvements:

1. We constructed two new *promoter::YFP-NLS* (a ribosome function-related gene and *THAS1*, Extended data Fig. 5b and Fig. 5c, g-h) transgenic reporter lines in the revised manuscript. This is significant because those transgenic lines provided more genetic data for validating our key conclusions from the snRNA-seq dataset.
2. We added new physiological data as suggested by reviewers. For example, we found that GMI1000 does not significantly induce cell death during the first 6 hours of interactions (Extended data Fig. 7). We also showed that *thas1* mutant does not affect the direct resistance to GMI1000, and thus the effect of THAS1 on microbiome re-shaping is due to its effect on microbiome structure rather than direct disease resistance.
3. We added new data analysis as suggested by reviewers. We revealed that several canonical immune marker genes also show maturation zone-specific enriched expression patterns in response to GMI1000 (Fig. 4e-g) and plotted the expression patterns of the IGS pathway (Extended data Fig. 9).
4. All microbiome and snRNA-seq data had been properly deposited in a publicly available database.

REVIEWER COMMENTS

Reviewer #1 (Remarks to the Author):

In this study, Yang, Li, and Guan used single-nucleus RNA-seq to investigate the responses of Arabidopsis roots colonized by either beneficial (*Pseudomonas simiae* WCS417) or pathogenic (*Ralstonia solanacearum* GMI1000) bacteria. Their data indicate that roots show both common and distinct responses to beneficial and pathogenic microbes. They found that genes related to ribosome complex formation, translation, are specifically induced by WCS417 in the proximal meristem. They also showed that GMI1000 can induce immune genes (especially glucosinolate-related genes) in mature epidermis, while WCS417 did not. The authors identified an induction of triterpene biosynthesis genes not observed in a previous bulk RNA-seq study and showed that one of these genes is required for root microbiome assembly, highlighting the utility of single cell analysis. Overall, this study provides new insights into cell type-specific responses of roots to colonization by bacteria with different lifestyles, with validation and some functional analysis.

The manuscript is clearly written, but some claims related to Fig. 6 require more attention. In Fig. 6, the authors explore the role of THAS1, which is induced in a cell type-specific manner upon GMI1000 infection. While this section suggests that THAS1 is involved in microbiome assembly and responses to GMI1000 infection, the data do not fully support the authors' conclusions. The authors seem to draw conclusions beyond the available data. Detailed comments are provided below.

Response: Thanks for all the help in improving our manuscript. We corrected the

mismatched ASV names in the revised version, and we re-wrote most the result and conclusion parts related to Fig. 6 to address the unclear and over-concluded points before.

==Major comments==

COMMENT-1

The manuscript lacks sufficient detail regarding the codes used for analyses and figure generation. To ensure reproducibility, it is important that all raw data and codes needed to replicate key analyses are made readily available. This includes detailed information about the analysis environments, such as the specific packages and versions used. Ideally, this should be provided in a containerized format using tools such as Docker. In addition, repositories for the code need more comprehensive documentation to effectively support the research community. Finally, the methods section of the manuscript should be expanded to provide more detail.

Response: Thanks for the helpful suggestion.

[1] We have put our codes for all snRNA-seq analysis in the Github website available for the community (https://github.com/ZhaiLab-SUSTech/snRNA-seq_microbes).

[2] We deposited all snRNA-seq and microbiome data in the China National Center for Bioinformatics database.

[3] We added a new paragraph to describe methods related to microbiome data analysis.

[4] We added more detailed information in the revised method part (in the tracking change model).

COMMENT-2

Fig3c-e is interesting. Can the authors check the colonization of bacteria in the mutants? Is the lack of growth suppression related to the impaired colonization or something downstream?

Response: Thanks for the helpful suggestion. We added a new Extended data Fig. 6 related to this comment:

Lines 266-270: “We also measured the colonization levels of WCS417 and found that there was no significant difference between mutants and Col-0. This suggests that *cdc123*, *hem1* mutants triggered incompatible interaction with WCS417 is not due to impaired colonization, which could be due to downstream events related to translational regulation (Extended data Fig. 6).”

COMMENT-3

Could the authors confirm whether DST5 and DST7 are expressed in proximal meristem? This information is important to link a cell type to function.

Response: Thanks for the comment. We agree that it would be interesting if *CDC123* (AT4G05440) and *HEM1* (AT2G35110) have tissue-specific expression patterns. However, considering their strong function in translational regulation, they should be essential for translation events broadly required for many cell types. Actually, at the transcriptome level, they both do not show strong cell type-specific expression patterns (see dot plot graph below).

Since we had provided solid genetic evidence that those two mutants affect the compatible interactions with WCS417, our result suggests that their roles in root-WCS417 interaction are not due to cell type-specific expression patterns.

COMMENT-4

Validation of ribosome related genes in space would be needed as this is a major finding of this study.

Response: Thanks for the helpful comment.

[1] As suggested, we constructed a transgenic reporter line using the promoter of ribosome-related gene *AT3G03600* to drive the expression of the YFP with a nuclear localization signal (NLS) reporter line (*pAT3G03600::YFP-NLS*). We used the confocal bio-imaging system to show that *AT3G03600* showed enriched expression in the proximal meristem cells, which experimentally validated the expression pattern “in space” detected by our snRNA-seq analysis (Extended data Fig. 5).

[2] We added new results and discussion related to this part (lines 243-256):

“We identified a total of 249 ribosomes function-related genes³³ in 1055 genes from cluster D. To validate their expression patterns in space, we used the root cell atlas website tool (<https://rootcellatlas.org>) to check gene expression patterns in a high-throughput way. This tool integrated diverse available public Arabidopsis root single-cell datasets^{22,34-39}, and provided high-confident gene expression patterns based on the studies from different labs. We selected the top 53 highest expressed genes from those 249 ribosome-related genes, and found that they all show perfectly enriched expression patterns in meristem cells (Extended data Fig. 5). This further confirms the consistency and robustness of our data with previous single-cell studies in roots. Importantly, we also constructed a *pAT3G03600::YFP-NLS* reporter line, and experimentally validated the meristem cells enriched expression in space by confocal bio-imaging analysis.”

COMMENT-5

In Fig 5g, the authors attempted to show atrichoblast/cortex-specific expression of THAS1 using a reporter line. However, the presented images are unclear and it is difficult to assess the cell type specificity. Please provide clearer data. Typically, a nuclear localization signal (NLS) reporter line provides a clearer signal of cell specificity, as the authors have done in Fig. 4h. The authors should also evaluate the inducibility of THAS1 by GMI1000 infection using the reporter line.

Response: Thanks for the suggestion, we agree that a nuclear localization signal (NLS) reporter line would help better reflect and illustrate the expression/responsive patterns. As suggested, we constructed a transgenic reporter line using *THAS1* promoter to drive the expression of the YFP with a nuclear localization signal (NLS) reporter line (*pTHAS1::YFP-NLS*). We got two new experiment results:

[1] Spatial expression: We used Z-stack based confocal bio-imaging approach to show that *THAS1* showed elevated expression in the cortex cells upon GMI1000 treatment, which experimentally validated the expression pattern detected in our snRNA-seq analysis (new Fig. 5c).

[2] Cell type-specific responsiveness: We used the *pTHAS1::YFP-NLS* reporter line to further validate the developmental stage-specific responsive patterns of *THAS1* to GMI1000 infection. Consistent with our snRNA-seq profiling, we observed a much stronger expression of the *pTHAS1::YFP-NLS* signal in the maturation zone compared to the very weak induction in the meristematic zones after GMI1000 infection (new Fig. 5g-h).

COMMENT-6

Fig 6e is confusing. It's hard to see what's being compared in each plot. A more detailed legend would be helpful. The fold change values for Col-0 seem to be incorrect.

Shouldn't the authors expect a positive value for Oxalobacteraceae, since this strain is enriched upon GMI infection? Also, the x-axis says GMI100, not GMI1000.

Response: Thanks for the comments. We re-made the graph and also carefully updated the legend to avoid confusion.

COMMENT-7

If THAS1 recruits beneficial strains upon GMI infection, then plant growth will be reduced in the *thas1* mutant upon GMI infection in the presence of the microbiome compared to WT. I suggest that the authors test this. Comparing WT and *thas1* with GMI alone might also be informative as this will tell us if there is a microbiome independent role of THAS1 in defense against GMI.

Response: Thanks for the cool suggestion!

[1] To study whether the *thas1* mutant has microbiome independent effect on GMI1000 resistance, we have to test GMI1000 resistance or infection in a gnotobiotic system. A previous method paper nicely addressed this requirement [1], which inoculates root tips with GMI1000 and detects the colonization levels of GMI1000 in shoots (see Method part in revised MS Lines 749-756). This helps accurately reflect the disease progression in different plant mutants because *Ralstonia* is a systematic vascular pathogen that usually attacks roots but could be detected systematically in shoot tissues. Based on this method, we found that the colonization level of GMI1000 shows no significant difference between Col-0 and *thas1* mutant on plates [in the absence of microbiome (New Extended data Fig. 10)]. Our data confirmed the effect of THAS1 on microbiome re-shaping upon GMI1000 infection is not due to the direct role on GMI1000 resistance (Lines 395-399).

[1] We understand that if *THAS1* recruits beneficial strains upon GMI1000 infection, then we should expect some detrimental plant phenotypic changes related to

microbiome changes in *thas1* upon GMI1000 infection. However, not all microbiome changes can be drastically reflected at the “fresh weight” phenotypic level, and the effect of subtle microbiome changes is usually extremely hard to be nicely reflected as drastic plant fresh weight differences. In the field, 16S rRNA based microbiome amplicon sequencing is the most powerful approach to statistically confirm the overall microbiome changes in different mutants [2, 3]. In this part, our key data is the microbiome composition analysis results, and we also tried to modify our conclusion to be a microbial ecological part rather than solid experimental confirmation. This is significant because, to our knowledge, this provided the first genetic evidence that GMI1000-induced microbiome re-shaping (based on beta diversity) is largely blocked in a single mutant *thas1*. In the future, we could conduct a metagenome-based microbiome functional analysis, to better reflect the effect of *thas1* on microbiome function changes upon GMI1000 infection.

COMMENT-8

The conclusions drawn from Fig 6f-h are not convincing. It is not very surprising that some commensals can protect plants from GMI1000. It will take a more comprehensive screening of commensals to convincingly say that commensals recruited by plants using the THAS1 pathway protect plants.

Response: Thanks for the critical comment! This part mainly provided sequencing-based microbial ecological evidence that THAS1 is genetically required for microbiome reshaping from a community composition level. However, it would be hard to match all detected ASVs to real isolates, and also extremely hard to “convincingly confirm that commensals recruited by plants using the THAS1 pathway protect plants”.

We tried to add discussion and modified our previous conclusion to fix this concern.

We agree that “a more comprehensive screening of commensals” would strengthen the biological relevance of microbiome changes. However, it is probably the most difficult challenge in microbial ecological studies, especially for rhizosphere microbiome studies (one of the most complicated microbial communities on this planet). It is always hard to isolate all microbes and match all interested ASVs from microbiome profiling data. In the field, it is usually hard to luckily match all interested ASVs from microbiome profiling and test their functions. For example, a previous study revealed the drastic effects of prolonged drought on rice root microbiome changes, but only perfectly matched and confirmed the stress-protecting effect of one strain (*Streptomyces* sp. SLBN-177, Nature Plants, 2021[4]). A similar study tried to match *Ralstonia* resistant microbes from resistant tomato cultivars, and only successfully matched one strain (Kwak et al, Nature Biotech, 2018 [5]).

We corrected our previous mistake in showing the mismatched ASV number in Sup Table 6, and substantially improved this paragraph from line 419 to 426. We added more explanation to avoid overstating our conclusion:

“However, since we only isolated 4 Oxalobacteraceae strains, whether this family is mainly responsible for disease suppression requires more strain isolation and functional validation in the future. Even though our root microbiome composition analysis confirmed that: a) GMI1000 infection triggers extensive root microbiome reshaping; b) this microbiome reshaping process is genetically blocked in *thas1* mutant.”

COMMENT-9

In Line 376, the authors claimed “This indicates that wild-type plants repel the colonization of Comamonadaceae strains during *Ralstonia* infection, and this effect is genetically dependent on THAS1.” Is there any evidence where plants “actively” repel Comamonadaceae during GMI1000 infection? It is possible that GMI1000 directly affects Comamonadaceae?

Response: Thanks for the comment, and we have updated this inaccurate expression:

“This indicates that wild-type plants might directly or indirectly suppress the colonization of Comamonadaceae strains during *Ralstonia* infection, while this effect is genetically dependent on *THAS1*.”

COMMENT-10

Appropriate statistical tests or replicate information are missing in some figures, such as Fig 3de and Extended Data Fig 1.

Response: Thanks for the comment. We have thoroughly checked and updated the figure legend in the revised version, especially related to statistical information following the checklist file (new Fig. 3d-e and Extended data Fig. 1).

==Minor comments==

COMMENT-11

Showing the number of cells per merged cluster (11 clusters) for each condition (mock, colonization) would be helpful.

Response: Thanks for the comment. We had provided a new Extended data Fig. 3 to illustrate the number of cells per merged cluster (11 clusters), as well as number of detected genes per cluster under each condition.

COMMENT-12

Fig 2c shows strong induction of auxin related genes in atrichoblast and cortex upon GMI1000 infection. Since the signal is very strong, I'd appreciate more discussion on the strain and cell type specific induction of this process.

Response: Thanks for the comment. We added more information in the result part:

“For GO terms mainly enriched in GMI1000-induced DEGs, we found that “response to auxin” is enriched in the highest number of cell types (7 of 11 annotated cell types). This is consistent with the previous studies that GMI1000 may produce auxin-like molecules³⁰, and induce the expression of DR5 gene (a marker gene of auxin signaling)³¹. GMI1000 might induce the expression of auxin biosynthesis, signaling, and transport genes to manipulate root architecture upon infection³².”

COMMENT-13

In Extended data Fig 3c, does the number of DEGs in each cell type after different treatments reflect the number of cells analyzed or the quality of the data, such as genes detected per cell (and thus statistical power)? This needs to be assessed to make a claim. “indicating that root proximal meristem cells display the most distinct responses to beneficial and pathogenic microbes. (Line 232)”

Response: Thanks for the comment. We have provided a new Extended data Fig. 3 to illustrate the number of detected genes per cluster is quite similar under each condition.

COMMENT-14

In Fig4, they used 970 core immune genes previously identified using seedling experiment. It is not specifically root immune genes, so focusing on these genes may bias the analysis. I suggest performing more comprehensive analysis to identify developmental stage specific DEGs followed by GO enrichment or overlap analysis against the core immune genes. This will be a similar analysis to Extended Data Fig 5 but with all genes.

Response: Thanks for the helpful comment!

[1] We conducted “a similar analysis to Extended Data Fig. 5 but with all genes” as suggested, however, in that case, we cannot detect many maturation cell specific responsive patterns due to too many groups and DEGs (see below heatmaps);

[2] To avoid bias, we also checked the expression of more general immune or SA-related genes. Notably, we observed that several immune or SA-related marker genes including *PBS3*, *MPK11*, *CBP60g*, and *SID2* show similar expression patterns with IGS pathway (strong induction in response to GMI1000 in maturation zone, new Fig. 4e-g).

This further confirms that there are also some canonic immune or SA-related marker responses that show strong induction in the maturation zone, which further confirms our previous findings (Lines 333-339).

***All DEGs from sub-clusters belonging to Cortex**

***All DEGs from sub-clusters belonging to Endodermis**

***All DEGs from sub-clusters belonging to Atrichoblast**

COMMENT-15

How do the IGS related genes look like in other cell types? Since the finding is interesting it would be informative to show the comprehensive picture of the expression of these genes in the atlas.

Response: Thanks for the suggestion. We have analyzed the IGS-related genes expression pattern in other cell types. They tend to be enriched in the cortex cell type, and also show some enrichment in the unknown cell types.

COMMENT-16

Fig 5b is unclear. It does not help showing THAS1 induction in GMI1000 treated roots.

Response: Thanks for the comment. We constructed a new *pTHAS1::YFP-NLS* reporter line, which was used to confirm not only the basal level cell-type specific enriched

expression pattern, but also the maturation cell enriched responsiveness to GMI1000 (New fig. 5c,g-h).

Reviewer #2 (Remarks to the Author):

This manuscript reports an interesting investigation of plant responses to both beneficial and pathogenic bacteria using single-nucleus RNA-seq. While the localized immune response in plant roots has been studied using transgenic reporters, our understanding of genome-wide transcriptional changes related to the spatial organization of immunity during bacterial colonization at single-cell resolution remains limited. This paper provides insight into the localized and cell-type specific pathways in plant roots in response to rhizosphere microbes. Although I am generally positive about the overall premise of the paper, there are several aspects regarding data analysis and experimental details that need to be clarified. I have outlined my comments below.

Major comments:

1) How is the expression of HEM1 and CDC123 in different cell types after different treatments? Do these two genes belong to cluster-D (1055)? Do pathogen treated cells show higher expression than the beneficial bacteria? Please provide data regarding the expression of HEM1 and CDC123 in different cell types after different treatments.

Response: Thanks for the comment. We agree that it would be interesting if *CDC123* (AT4G05440) and *HEM1*(AT2G35110) have tissue-specific expression patterns. However, considering their strong function in immune-regulated translational regulation, they should be essential for translation events broadly required for many cell types. Actually, at the transcriptome level, they both do not show strong cell type-specific expression patterns (see dot plot graph below).

Since we had provided solid genetic evidence that those two mutants affect the compatible interactions with WCS417, our result suggests that the role of those translational regulators in root-WCS417 interaction is not due to cell type-specific expression patterns.

2) How many core DEGs are overlapped in cluster 11, 17_0, 19, 17_1? Compare the expression level of the overlapped genes in these clusters.

Response: Thanks for the comment. There are 472 core DEGs in these clusters (Both Up and Down-regulated genes were used). These overlapping DEGs also show developmental stage-specific responsive patterns. However, currently, our manuscript does not have conclusions related to the overlap between cortex and atrichoblast cells, we did not include this result in the revised manuscript.

3) Lines 280-281, the meristem shows the largest response to WCS417, while maturation zone shows the most significant response for GMI1000. This might be due to the different interaction/infection strategies of two bacteria, as *Ralstonia* prefers to infect plants through wounds and lateral root emergence sites. Therefore, the different immune strategies that root employed at the very early stage of the interaction may result from the colonization/infection preference of various bacteria.

Response: Thanks for really interesting comment. We added new discussion related to the idea that “Ralstonia prefers to infect plants through wounds and lateral root emergence sites” in line 522-525. For WCS417, we currently do not have evidence to show that it shows enriched colonization in root meristem cells, and was not over-concluded here.

4) Lines 282-288, It is true that maturation/differentiation zone (atrachoblast cells) loses responsiveness to pure elicitor treatment, but it shows strong response when coupled with damage (Zhou et al., 2020). Thus, maturation/differentiation zone mounts an immune response to pathogen infection due to coincidence of MAMPs and damage. The section between lines 282-288 gives the impression that the immune response of the maturation/differentiation zone was previously unknown. Please edit this section.

Response: We apologize for the inaccurate data interpretation before.

[1] At the beginning of the paragraph, we added a clear description: “Plant roots exhibit cell-type and developmental stage-specific responses to various immune elicitors. A significant finding is that mature atrichoblast cells largely lose their immune responsiveness to pure elicitors, and can only mount immune responsiveness in the coincidence of damage and elicitor. However, whether root maturation zone still responds to live pathogen or beneficial microbe is not clear, especially from a cell type-specific transcriptome perspective.”

[2] At the end of that paragraph, we added a new result (lines 298-305) to show that GMI1000 does not induce significant cell death in the maturation zone.

5) Elongation zone doesn't show expression in figure 4 h-I, but lines 310-311 mention strong expression. Please ensure consistency between the figure and the text.

Response: Thank you for pointing out this mistake. We deleted the “elongation zone” in that part in revised version.

6) It is interesting to see highly specific induction of CYP71A12 and THAS1 genes in the maturation zone. Do other defense related secondary metabolite genes (both snRNA-seq data and transgenic lines) show this pattern? How about the general immune related marker genes, e.g. FRK1, PER5, MPK11, WRKY33, SA related genes etc.? Do they show the same or different pattern?

Response: Thanks for this helpful comment, we added new data based on this suggestion.

[1] We checked the expression patterns of genes related to other root microbiome interaction-related secondary metabolic pathways, including genes related to scopoletin and flavonoid biosynthesis. However, we did not see a maturation cells-specific response for those pathways (Extended Data Fig. 12). That indicates root maturation

cells show specificity for the induction of phytoalexin and thalianol biosynthesis genes upon *Ralstonia* infection.

[2] We also checked whether general immune or SA-related genes show similar expression patterns to *CYP71A12*. Notably, we observed that several immune or SA-related marker genes including *PBS3*, *MPK11*, *CBP60g*, and *SID2* show similar expression patterns (strong induction in response to GMI1000 in maturation zone, new Fig. 4e-g). This further confirms that there are also some canonic immune or SA-related marker responses that show strong induction in the maturation zone, which further confirms our previous findings (Lines 333-339).

7) THAS1-YFP localization is interesting. It seems to co-localize with cell wall/plasma membrane or outside of cells. Please include a colocalization assay to confirm THAS1-YFP is on the cell wall/plasma membrane of atrichoblast and cortex cells but not outside of cells. A transcriptional reporter (pTHAS1-YFP) might also directly demonstrate its expression location.

Response: Thanks for the insightful comment.

[1] As suggested, we constructed a transgenic reporter line using *THAS1* promoter to drive the expression of the YFP with a nuclear localization signal (NLS) reporter line (*pTHAS1::YFP-NLS*). We used a confocal bio-imaging system to show that *THAS1* showed elevated expression in the cortex cells upon GMI1000 treatment, which experimentally validated the expression pattern detected by our snRNA-seq analysis (see new Fig. 5).

[2] We also used the *pTHAS1::YFP-NLS* reporter line to validate the developmental stage-specific responsive patterns of *THAS1* in response to GMI1000 infection. Consistent with our snRNA-seq profiling, we observed a much stronger induction of the *pTHAS1::YFP-NLS* signal in the maturation and elongation zones compared to the very weak expression in the meristem zone after GMI1000 infection (new Fig. 5).

8) Lines 395-400, what about GMI1000 infection and growth in this process? It would be interesting to investigate whether GMI1000 can still infect plants or if 22G3/10G7 has the ability to kill or outcompete *Ralstonia*. Please check and compare the population levels of GMI1000 and 22G3/10G7.

Response: Thanks for the comment, we agree that how commensal strains exhibit biocontrol activity is interesting. But so far we still know little about the exact biocontrol mechanisms during friend/foe co-colonization. We tested the GMI1000 colonization levels when co-inoculated with 10G7 and 22G3. However, 22G3 co-inoculation has no significant effects on GMI1000 root colonization level, while 10G7 only has a slight effect on GMI1000 growth on the root surface. This suggests that direct

pathogen out-competing is not the major reason for biocontrol activity observed in these two strains.

Fig description: We utilized antibiotic resistance to distinguish the GMI1000 and 22G3/10G7. We transferred the GMI1000 strain with a gentamicin resistance vector (renamed GMI1000g), while 22G3/10G7 has no gentamicin resistance. Interestingly, we found that 10G7 only slightly inhibits the colonization of GMI1000g, while 22G3 cannot inhibit the population of GMI1000g. Those suggest that commensal outcompeting cannot fully explain the biocontrol ability.

9) Lines 387-397, “three strains (9C5, 1H4, 22G3) exhibited high 16S rRNA similarity to ASV9 (Supplementary Table 6)”. “... both 22G3 (ASV9-like) and 10G7 (ASV5-like) strains”. These descriptions are not consistent with Supplemental table 6, which shows 9C5, 1H4, 22G3 are ASV12 or ASV20-like. Please check and provide a justification.

Response: Thanks for pointing out the mistake. We checked Supplemental table 6 and corrected it in the revised version, we re-wrote the whole microbiome part related to strain function validations (line 421-452).

10) Lines 438-439, GMI1000 induces cell death at 12 hpi but not at 6 hpi (Zhou et al., 2020). The data in this manuscript was obtained at 6 hpi. The bacterial ODs were different: 0.1 in Zhou et al., 2020 and 0.05 in this MS. Does the single-cell data in this

manuscript show cell death related pathways at 6 hpi? Or the authors might experimentally check root cell death at 6 hpi. I wonder if the immune response triggered at 6 hpi is cell death/damage dependent or not?

Response: Thanks for the helpful comment, we agree that matching this physiological information will deepen the understanding of the immunological basis of our molecular observations.

[1] We used PI staining-based imaging system to check whether GMI1000 induces cell death in the maturation zone. We found that does not induce strong cell death (broad PI staining inside the cell [6]) in the maturation zone.

[2] Does the single-cell data in this manuscript show cell death related pathways at 6 hpi?

We checked the expression patterns of cell death-related genes from GO: 0008219. We did not detect significant and strong cell type-specific expression patterns (dotplot below).

Minor comments:

1) Line 106, delete “(GMI1000 hereafter)”

Response: Thanks for the comment. We had deleted “(GMI1000 hereafter)” in the revised version.

2) Lines 179-180, rephrase this sentence “we also observed induction of two PSKR1 ligand genes, Phytosulfokine (PSK) 1 but not PSK2,”. It’s confusing if PSK2 is induced or not.

Response: Thank you very much for pointing out this mistake. We have corrected it in the manuscript accordingly. We rephrased the sentence to “We also checked the induction of two PSKR1 ligand genes (*Phytosulfokine (PSK) 1* but not *PSK2*). We found that only *PSK1* was induced by beneficial WCS417 but not GMI1000 (Extended Data Fig. 4a).”

3) Line 563, correct the sentence “and the 517-617 nm” method

Response: Thanks for the comment. We have corrected the sentence “and the 517-617 nm” with “YFP was excited at 514 nm”.

4) Line 823 and 827, duplicate reference

Response: Thank you for pointing out this mistake. We have thoroughly checked our references and updated them accordingly.

5) Line 831 and 833, duplicate reference

Response: Thank you for pointing out this mistake. We have corrected them in the manuscript accordingly. We deleted the duplicate reference and updated it in the revised version.

6) Figure 6e, “***” in the legend does not appear in figure. Please provide statistical method here.

Response: Thank you very much for pointing out this mistake. We have corrected them in the manuscript accordingly. We deleted the “***” and updated it in the revised version.

7) Extended data Figure 1 legend is incorrect. Please edit.

Response: Thank you for pointing out this mistake. We have corrected them in the manuscript accordingly.

References

1. Yu, G., et al., *Inoculation of Arabidopsis seedlings with Ralstonia solanacearum in sterile agar plates*. STAR Protoc, 2023. **4**(3): p. 102474.
2. Chen, T., et al., *A plant genetic network for preventing dysbiosis in the phyllosphere*. Nature, 2020.
3. Lebeis, S.L., et al., *Salicylic acid modulates colonization of the root microbiome by specific bacterial taxa*. Science, 2015. **349**(6250): p. 860-4.
4. Santos-Medellín, C., et al., *Prolonged drought imparts lasting compositional changes to the rice root microbiome*. Nature Plants, 2021. **7**(8): p. 1065-1077.
5. Kwak, M.-J., et al., *Rhizosphere microbiome structure alters to enable wilt resistance in tomato*. Nature biotechnology, 2018. **36**(11): p. 1100-1109.
6. Zhou, F., et al., *Co-occurrence of Damage and Microbial Patterns Controls Localized Immune Responses in Roots*. Cell, 2020. **180**(3): p. 440-453 e18.

Reviewer #1 (Remarks to the Author):

I appreciate the authors' efforts in addressing my previous comments. While many of my concerns have been resolved, several major issues remain. A key shortcoming of the manuscript, in my opinion, is limited compelling evidence that links cell type-specific gene expression changes to root-bacteria interactions. This gap significantly limits the impact of the study, as it does not convincingly demonstrate that the snRNA-seq analysis has led to new mechanistic insights, as implied by the title and abstract. Below are my detailed comments.

Reply-1: Thanks for all the helpful comments on this manuscript, and for catching this critical logical gap! We believe this “gap” arises because we did not clearly state the strong links between CDC123/HEM1 and the “cell type-specific gene” in the last version (as we discussed below and in the manuscript).

We had fully addressed this key comment in two directions:

1) We added new statements to clearly explain that CDC123/HEM1's target proteins belong to the WCS417-induced cluster D genes (**Line 262-271**);

2) We have conducted new experiments and provided new genetic data by testing the phenotype of 13 more ribosome related mutants directly derived from WCS417-responsive gene clusters identified in our snRNA-seq data (cluster D genes, New Fig. 3f-g). Those are substantial amount of new solid genetic results and perfectly provide “compelling evidence that links cell type-specific gene expression changes to root-bacteria interactions”, as requested by Reviewer #1.

Our new genetic validations, along with the added statements in the manuscript, comprehensively address this major comment from two angles.

Revision Comment 1

As both initially and currently written, the manuscript implies that HEM1 and CDC123 are specific to the proximal meristem, and their specific expression may play an important role in WCS417-mediated plant growth promotion, which would have been an interesting finding. However, in response to my previous Comment 3, the authors have shown that CDC123 and HEM1 do not exhibit cell type specificity. This result has led me to revise my initial impression and reassess the study. A significant weakness of the manuscript is the lack of data supporting the role of cell type-specific gene induction in the beneficial effects of WCS417. As it stands, the manuscript feels like a combination of an snRNA-seq dataset and unrelated mutant analyses. Although the authors demonstrated that *Thas1* expression is cell type-specific, this gene is already known to contribute to plant-microbe interactions, limiting the impact of the finding. I recommend focusing on genes that display cell type-specific expression changes in response to WCS417 and further investigating their roles in promoting beneficial plant-microbe interactions.

Reply-1: Thanks for this critical and helpful comments. We tried to fully address this concern in two different directions: a) we tested 13 more mutants from WCS417-induced cell type-specific up-regulated ribosome-related genes; 2) We provided further clarification of the direct link between HEM1/CDC123 and WCS417-induced cell type-specific up-regulation of ribosome related genes.

1. We obtained 13 more mutants related to ribosome assembly from WCS417-induced cell-type specific up-regulated ribosome genes in proximal meristem cells (cluster D), with 6 of them also dampening WCS417-induced lateral root growth promotion (New Fig. 3f-g). This new solid genetic evidence suggests that there are diverse ribosome assembly-related genes that “specific to the proximal meristem, and their specific expression plays an important role in WCS417-mediated plant growth promotion” —precisely the “interesting finding” that Reviewer 1 suggested;
2. We added new statements to clarify the link between HEM1/CDC123 and WCS417-induced ribosome genes in proximal meristem cells (**Line 262-271**), demonstrating that our findings are not merely a combination of an snRNA-seq dataset and unrelated mutant analyses.

Line 262-271:

“HEM1 had been reported to be a master upstream hub regulator for organizing translation components, which interacts with 35 cytosolic ribosome translation initiation, elongation or release factors[38]. Among those 35 ribosome proteins, 12 of them belong to the WCS417-induced cell type-specific DEGs in our cluster D (Extended Data Fig. 6). CDC123 regulates the assembly of eukaryotic initiation factor 2 (eIF2) heterotrimer (α , β , and γ subunits) complex, which is central for translation initiation [39]. CDC123 protein can be immune-precipitated with 13 translation initiation factors ([39]), among them 6 belongs to WCS417 specifically induced DEGs from cluster-D (Extended Data Fig. 6a and Extended Data Fig. 6b).”

Meanwhile, we also discussed why the expression of HEM1 and CDC123 seems not under tight transcriptional regulation (Extended data Fig. 6). It might be because that those two master upstream translational regulators are mainly regulated at the post-transcriptional level. For example, CDC123 has an ATP grasping domain and its translational regulating activity is gated by ATP concentration (Extended Data Fig. 6a), and HEM1’s activity is also regulated at the post-translational level by a phase separation mechanism to condensates with translation components(Zhou et al., 2023). We had properly added this explanation in our revised manuscript.

Additionally, we tried to response the significance of our THAS1-related microbiome results:

* Although THAS1 has been reported to regulate root microbiome composition under normal growth conditions [<https://www.science.org/doi/10.1126/science.aau6389>], the biological context in which this pathway works to sculpt the microbiome remains elusive. This is an important question because understanding the ecological and evolutionary relevance of this pathway in shaping the microbiome is essential. In the plant microbiome field, many mutants can directly or indirectly shift microbiome composition, but establishing the biological relevance of these changes is challenging (reviewed in <https://onlinelibrary.wiley.com/doi/10.1002/imt2.8>). This is probably because that the 16S rRNA-based molecular detecting approach is highly sensitive, many metabolic or stress-related genes might indirectly affect diverse processes and thereby shift microbiome structure. In this work, we observed that *thas1* mutant blocked the statistically significant microbiome beta-diversity changes that occurred in wild type plants (Fig. 6C). This is a striking result for microbiome-related phenotypes, because for now we haven’t seen (from literature) a single plant gene mutation blocking such statistically significant microbiome beta diversity changes induced by *Ralstonia* infection. Furthermore, our lab has isolated more than 1500 root microbe strains and characterized 256

different microbes (Another independent work which will be online published this or next month). This collection allowed us to match some differentially abundant microbes like Oxalobacteraceae strains and test their biocontrol activities (a reductionist approach considered the most robust method to explore microbiome functionality <https://www.sciencedirect.com/science/article/pii/S1369527419300414?via%3Dihub>). Our microbiome statistical analysis and the further phenotypic validation of key taxa's (Oxalobacteraceae) phenotypes represent a standard microbiome story.

*Lu et al. (2018) reported that DR5 expression is induced in the vasculature upon GMI1000 infection, but not in atrichoblast or cortex cells, which seems to contradict the results of the snRNA-seq analysis presented here. How do the authors reconcile this inconsistency? This also raises concerns about the extent to which the snRNA-seq data has contributed to our understanding of plant cell type-specific responses that govern root-microbiome interactions.

Reply-2: Thanks for the very careful checking. We would like to clarify the following:

- 1) There are substantial differences (as listed below) in the experimental setup between our work and that of Lu et al. (2018). For the expression patterns of auxin responsive reporter line, it would be affected by many different experimental factors such as plant age, growth, and treatment systems;
- 2) There are also previous study that shows similar trends with our data (<https://academic.oup.com/plcell/article/31/8/1767/5985761?login=true>) (Jing et al., 2019);
- 3) We conducted new experiments to confirm that DR5:GUS signal is presented in the cortex and atrichoblast cells.

[1] There are substantial differences (as listed below) in the experimental setup between our work and Lu et al. (2018) (Lu et al., 2018).

Experimental setup differences between our work and Lu et al. (2018)		
Different factors	Our work	Lu et al. (2018)
Plant age	12 days after germination	Six-day-old
GMI1000 treatment peroid	6 hours	24 and 48 hours
Growth condition	48 well-plate based hydroponically plant-rhizosphere microbe interacting system (Haney et al., 2015), plant roots are exposed to liquid medium.	Vertical MS plates, plants are grown on solid surface.

[2] There are also previous study which shows similar trends with our data. In previous paper (Fig 5-C,D) from Prof. Sheng Luan's lab (<https://academic.oup.com/plcell/article/31/8/1767/5985761?login=true>) (Jing et al., 2019), they confirmed that root immune activation can rapidly induce DR5 expression both in the stele and surface cell types [including atrichoblast (epidermal) or cortex cells, see Figs below from their paper]. It seems that DR5:GFP signal tends to be weaker than DR5:GUS signal in the atrichoblast cells, which might also explain why Lu et al. (2018) did not see very strong DR5:GFP

in the atrichoblast cells.

C. The DR5:GUS signal from seedlings incubated on half-strength MS agar medium supplemented with 100 nM Pep1 for 2, 4, 8, 12, and 24 h.

D. DR5:GFP signal from seedlings with Pep1 treatment for 1, 2 4, 8, 12, and 24 h.

[3] We experimentally validated the expression pattern of the DR5:GUS transgenic reporter line in response to GMI1000 after 6 hours inoculation. DR5:GUS signal was clearly observed in the “atrachoblast or cortex cells”. However, likely due to the damage or cell wall loosening enzymes caused by *Ralstonia*, most *Ralstonia*-treated root tips were broken after several more hours of GUS staining. This does not affect our conclusion that “DR5:GUS signal can be induced by *Ralstonia* in the “atrachoblast or cortex cells”. This further strengthens the robustness and accuracy of our snRNA-seq dataset.

Reply Fig-3: *Ralstonia* treatment (Six hours) induces DR5:GUS signal in both stele and atrichoblast or cortex cells

[4] We understand that the role of auxin in root-pathogen interactions is interesting and important. We will try to keep an eye on this direction in the next studies, however, more mechanistic studies are beyond the scope of this current work.

* In response to the comment regarding the extent to which the snRNA-seq data has contributed to our understanding of plant cell type-specific responses that govern root-microbiome interactions, our work makes several key contributions:

1) Our digital data and genetic validations confirmed that beneficial WCS417 broadly induces the expression of ribosome function-related genes in the meristem zone, and diverse WCS417-induced ribosome function-related genes are necessary for beneficial WCS417-induced growth promotion effects. This novel finding reveals a previously unreported critical role of ribosome proteins and translational regulators in the compatible interaction with beneficial microbes.

2) Our digital expression data and genetic validations confirmed that root maturation cells can mount immune response to live *Ralstonia* pathogens. This data extended the previous study which showed that root maturation cells cannot mount immune response to pure elicitors without damage

(Zhou et al., 2020). Compared to the previous Cell paper which used a limited number of immune-responsive transgenic marker lines to study root immune response, our dataset offers a comprehensive view of cell type-specific transcriptome responses to friends and foes;

3) Our expression data and genetic validations confirmed that the secondary metabolite triterpene biosynthesis-related gene *THAS1* exhibits cell-type-specific responses to *Ralstonia* infection, with a striking microbiome effect that *thas1* mutant cannot significantly shape its root microbiome in response to *Ralstonia* infection. We further conducted reductionist-based microbes isolation and functional validating experiments (which is currently the golden standard to confirm the biological relevance of microbiome sequencing results in either animal or plant microbiome field). This work addresses a critical question about the biological relevance of THAS1-mediated microbiome sculpting in plants, which was not answered in the previous Science paper (Huang et al., 2019).

4) Our study provided high accuracy (we validated most of our cell type-specific expression trends using marker line genetic evidence) and probably the first comprehensive snRNA-seq dataset for comparing root interactions with beneficial and pathogenic microbes. Importantly, It is value for root-microbe interaction studies, and two labs working on the *Ralstonia* pathogen had been interested in this work and requested the original dataset from our lab (because we previously posted a pre-print version).

Our snRNA-seq data had already successfully guided ongoing research in our lab too, leading to the identification of at least five additional regulators essential for root interactions with either *Ralstonia* or WCS417. Our data offers a valuable resource for identifying novel regulators of root-microbe interactions and microbiome structure.

Revision Comment 3

Extended data Fig. 1 is still missing statistical tests to claim significant gene expression changes.

Reply: Thanks for reminding, we added significance results for that.

Reviewer #2 (Remarks to the Author):

Authors addressed all my comments but one. Please address this concern.

Reply: Thanks for reminding, we fully addressed this concern from two directions.

1) Both CDC123 (AT4G05440) and HEM1(AT2G35110) do not exhibit specific induction in proximal meristem cells. Additionally, their induced expression appears to be nonspecific to WCS417, as high expression levels are observed in root cap, cortex, endodermis, procambium, pericycle, phloem and xylem etc. in response to GMI1000. Since 1,055 genes in Cluster-D are identified based on their highest induction in proximal meristem cells only after WCS417 but not GMI1000 treatment, this suggests that CDC123 (AT4G05440) and HEM1(AT2G35110) are not part of 1055 genes. Consequently, there is a disconnect between the session “beneficial microbe specifically induces the expression of translation-related genes in the proximal meristem cells” and the experimental validation of “two translation regulators (CDC123, HEM1)”. It seems that these two translation regulators are not relevant to the snRNA-seq data.

Reply-1: Thanks for catching this critical gap. We tried to fully address this concern in two

directions:

1) We obtained 13 more mutants related to ribosome assembly from WCS417-induced cell-type specific up-regulated ribosome genes in proximal meristem cells. Among these 6 of them were found to dampen WCS417-induced lateral root growth promotion (New Fig. 3f-g). This new solid genetic evidence established the solid connection between “beneficial microbe specifically induces the expression of translation-related genes in the proximal meristem cells” and the their role in the compatible interaction with beneficial microbes.

2) We added new model figures and statements to clarify the link between HEM1/CDC123 and WCS417-induced ribosome genes in proximal meristem cells, showing that our findings are not simply a combination of an snRNA-seq dataset and unrelated mutant analyses. HEM1 and CDC123 have been reported to directly interact with and regulate the activity of translation apparatus-related genes. Among 35 well-characterized direct downstream interacting partners of HEM1, 12 belong to our WCS417 up-regulated genes in proximal meristem cells. Additionally, CDC123 protein can be immune-precipitated with 13 translation initiation factors ([39]), among them 6 belongs to WCS417 specifically induced DEGs from cluster-D (Extended Data Fig. 6a and Extended Data Fig. 6b). So although “HEM1 and CDC123 are not specific to the proximal meristem”, their direct downstream targets related to ribosome functions show cell type-specific induction in response to WCS417. The expression level of HEM1 and CDC123 seems not under tight transcriptional regulation (Extended data Fig. 6). It might be because that those two master upstream translational regulators are mainly regulated at the post-transcriptional level. For example, CDC123 has an ATP grasping domain and its translational regulating activity is gated by ATP concentration (Extended Data Fig. 6a), and HEM1’s activity is also regulated at the post-translational level by a phase separation mechanism to condensates with translation components(Zhou et al., 2023). We had properly added this explanation in our revised manuscript.

We thank for this critical comment which guided us to make our claim more solid.

References

- Haney, C.H., Samuel, B.S., Bush, J., and Ausubel, F.M. (2015). Associations with rhizosphere bacteria can confer an adaptive advantage to plants. *Nature plants* 1, 15051.
- Huang, A.C., Jiang, T., Liu, Y.X., Bai, Y.C., Reed, J., Qu, B., Goossens, A., Nutzmann, H.W., Bai, Y., and Osbourn, A. (2019). A specialized metabolic network selectively modulates Arabidopsis root microbiota. *Science (New York, N.Y.)* 364.
- Jing, Y., Zheng, X., Zhang, D., Shen, N., Wang, Y., Yang, L., Fu, A., Shi, J., Zhao, F., Lan, W., and Luan, S. (2019). Danger-Associated Peptides Interact with PIN-Dependent Local Auxin Distribution to Inhibit Root Growth in Arabidopsis. *The Plant cell* 31, 1767-1787.
- Lu, H., Lema A, S., Planas-Marquès, M., Alonso-Díaz, A., Valls, M., and Coll, N.S. (2018). Type III Secretion - Dependent and -Independent Phenotypes Caused by *Ralstonia solanacearum* in Arabidopsis Roots. *Molecular Plant-Microbe Interactions* 31, 175-184.
- Zhou, F., Emonet, A., Denervaud Tendon, V., Marhavy, P., Wu, D., Lahaye, T., and

- Geldner, N.** (2020). Co-occurrence of Damage and Microbial Patterns Controls Localized Immune Responses in Roots. *Cell* **180**, 440–453 e418.
- Zhou, Y., Niu, R., Tang, Z., Mou, R., Wang, Z., Zhu, S., Yang, H., Ding, P., and Xu, G.** (2023). Plant HEM1 specifies a condensation domain to control immune gene translation. *Nature plants* **9**, 289–301.

REVIEWERS' COMMENTS

Reviewer #1 (Remarks to the Author):

I thank the authors for sufficiently addressing my concerns. I have no further major issues. However, I suggest that the T-DNA lines used in the revision be clearly described in the Methods section.

Thanks for the suggestion, we put all T-DNA mutant ID numbers into the Method part.

Reviewer #2 (Remarks to the Author):

Authors addressed my comment.

Thanks for all the help!